# Promoter G-quadruplexes and transcription factors cooperate to shape the cell type-specific transcriptome

Sara Lago [1✉], Matteo Nadai [1], Filippo M. Cernilogar [2], Maryam Kazerani [2], Helena Domíniguez Moreno[2], Gunnar Schotta [2✉] & Sara N. Richter [1✉]

Cell identity is maintained by activation of cell-specific gene programs, regulated by epigenetic marks, transcription factors and chromatin organization. DNA G-quadruplex (G4)-folded regions in cells were reported to be associated with either increased or decreased transcriptional activity. By G4-ChIP-seq/RNA-seq analysis on liposarcoma cells we confirmed that G4s in promoters are invariably associated with high transcription levels in open chromatin. Comparing G4 presence, location and transcript levels in liposarcoma cells to available data on keratinocytes, we showed that the same promoter sequences of the same genes in the two cell lines had different G4-folding state: high transcript levels consistently associated with G4-folding. Transcription factors AP-1 and SP1, whose binding sites were the most significantly represented in G4-folded sequences, coimmunoprecipitated with their G4-folded promoters. Thus, G4s and their associated transcription factors cooperate to determine cell-specific transcriptional programs, making G4s to strongly emerge as new epigenetic regulators of the transcription machinery.

[1] Department of Molecular Medicine, University of Padua, Padua, Italy. [2] Division of Molecular Biology, Biomedical Center, Faculty of Medicine, LMU Munich, Martinsried, Germany. ✉email: sara.lago@unitn.it; gunnar.schotta@med.uni-muenchen.de; sara.richter@unipd.it

Single-stranded nucleic acids that are rich in repeated guanine (G) tracts can fold into G-quadruplexes (G4s), noncanonical secondary structures. Stacked planar quartets stabilized by Hoogsteen hydrogen bonds connecting Gs of adjacent G-tracts form the G4 core[1]. Computational analysis of regular G4 sequence pattern predicted the abundance of putative G4s (pG4s) in genomic regulatory regions, such as oncogene promoters, splice and recombination sites, and telomeric ends[2–6]. Experimental analysis has in particular investigated and reported the implication of DNA G4s in the regulation of transcription: G4s were initially shown to represent physical obstacles to RNA polymerase processivity in vitro[7–13]; vice versa in the cellular environment using selective antibodies[14–18], chemical probing[19–22], and ligands[23–28], G4s were mapped and correlated to high gene expression[19,29]. Recent accumulating evidence suggests more complex functions of G4s in regulating transcription, as their effect depends on several factors: strand harboring the G4 (template or coding strand)[30], recruitment of transcription factors (TFs)[31–33], sequestering of chromatin remodeling proteins[21], and involvement in the distal genomic loci interaction[34]. Information on how G4s are distributed and modulated in different cell types, which would deepen our understanding of G4 functions in cells, is not currently available.

In the present work, we performed ChIP-seq analysis with a G4 antibody to map the folded G4s in cells of well-differentiated liposarcoma (WDLPS), a malignant neoplasia affecting the general human population, including young adults and children. WDLPS is resistant to current chemotherapies and has a largely unexplored biological background[35–37]. The integration of RNA-seq and assay for transposase-accessible chromatin followed by high-throughput sequencing (ATAC-seq) data allowed us to build a relationship between folded G4s, chromatin state, and transcriptional output. We also investigated G4 recruiting/displacement of TFs whose binding sites are enriched at G4 loci. Comparing our data with previously published G4 datasets and pan-cancer genomic variations, we were able to define for the first-time involvement of G4s in cell type identity determination.

## Results

G4s are dynamic structures that can be induced or unfolded upon interaction with G4-binding proteins[38–41]. We reasoned that the G4 landscape could be unique to each transcriptional program, hence cell type. We thus compared the G4 landscape in two cell types, and explored the specific interactions at the G4-folded and transcriptionally active sequences to ascertain the mechanism behind G4-mediated transcription regulation.

We investigated by G4-ChIP-seq[14] the in vivo DNA G4 folding in a cancer cell line of human WDLPS (93T449 cells), and compared data to those previously reported for the spontaneously immortalized keratinocyte HaCaT cells, from adult human skin[19]. These cell lines represent mesenchymal and epithelial differentiated cell types, respectively, the intrinsically different epigenetic/transcriptional background of which was exploited to assess genome-wide differences in the G4 landscape. To determine the G4 landscape in 93T449 cells, we purified BG4 antibody and in vitro validated its ability to discriminate G4 structures over non-folded single-stranded sequences by electrophoretic mobility shift assay (EMSA; Supplementary Fig. 1A)[42,43]. G4-ChIP in WDLPS cells was first validated by quantitative real-time PCR (qPCR) on control sequences (Supplementary Fig. 1B), then subjected to next-generation sequencing yielding ~5000 peaks that described the WDLPS genome-wide G4 landscape (Supplementary Data 1). Peak calling was performed using Homer software, which yielded comparable results to MACS2, with 86% and 89% of peaks identified by both tools for 93T449 and

HaCaT cells, respectively, but with Homer allowing the confident identification of a higher number of peaks (Supplementary Fig. 1C, D). GC content of 93T449 cells was significantly enriched in ChIP sample (~55%) vs the whole genome[44] (~41%), with a $T$ test $p$ value < 0.001; correspondingly, AT bases were significantly underrepresented (Supplementary Fig. 1E), with the highest GC frequency increasing toward the ChIP peak center, with a maximum GC occurrence at ~50 bp from the peak center (Supplementary Fig. 1F). The presence of G4s within ChIP peaks was assessed through Quadparser[3] and G4Hunter[45], and the predicted putative G4s referred to as pG4s. The mean length of the 7400–10,000 identified G4s was 21–36 bp (Supplementary Table 1), corresponding to that of the most stable G4s in vitro[46]. An average of 77% ChIP peaks contained at least one predicted G4 (Supplementary Fig. 2A, B), most contained 1–6 G4s, reaching 15 G4s in some sequences, with equivalent distribution between DNA strands (Supplementary Fig. 2C, D and Supplementary Table 1). Nonrandom distribution of G4s in the human genome has been previously reported[47]. In line with this, ChIP-G4s in 93T449 cells were strikingly prevalent in promoter regions (79%; Fig. 1A) with a strong enrichment of promoter elements in G4-ChIP peaks with respect to their representation in the whole human genome (Supplementary Fig. 4A). The second category in which G4s are most represented and positively enriched were 5′ UTRs (Fig. 1A and Supplementary Fig. 4A). 5′UTRs normally span from the gene TSS to the first exon ATG and partially overlap with promoters. In addition, TF-binding sites (TFBSs), other regulatory elements, splice sites, and structural transcription modulators are present within 5′UTRs[48,49]. Together, this evidence strongly supports G4 involvement in the transcription regulation. RNA-seq data (Supplementary Data 2) and subsequent analysis of the percentage of expressed genes that contained at least one ChIP-G4 vs G4 location showed that (i) G4-containing genes were more prone to be actively expressed with respect to G4-depleted genes; (ii) genes with G4s in promoter regions (annotated as promoter-TSS and 5′UTR) were almost 100% expressed (Fig. 1B); and (iii) produced significantly higher amounts of transcripts ($p$ value < 0.001; Fig. 1C). In particular, there was positive correlation between higher gene expression and the presence of G4s (Supplementary Fig. 4B), especially when G4s were within 1 kb from the TSS; this trend was smoothened as the distance from the TSS increased (Fig. 1D, top). Moreover, G4s mostly clustered within 250 bp from the TSS and the closer they were to the TSS, the higher the expression of the corresponding gene (Fig. 1D, bottom). Representative regions showing the relative position of G4s with respect to gene TSS are reported in Fig. 1E.

We next compared the amount of ChIP-G4s to that of pG4s computationally calculated using Quadparser (loops 0–12) and of "observed G4s" (oG4s), i.e., genomic regions previously observed to stop polymerase progression in vitro in the presence of $K^+$ (ref. [50]). From now on, pG4s refer to Quadparser- (loops 0–12) predicted G4; Quadparser was preferred over the other employed G4 prediction tools since it predicted the highest number of pG4s. Almost all ChIP-G4s were also found as pG4s (~100%) and oG4s (~95%) either when considering all genes or genes with G4s elsewhere than the promoter; in genes with G4s in their promoter, a similar percentage was found for ChIP-G4s matching pG4s (~100%), while a much lower percentage was found for oG4s (~23%; Supplementary Fig. 3A–D). Similar percentages were found performing the same analysis on HaCaT cell line (Supplementary Fig. 3A–D). The bias in pG4s and oG4s overlapping can be partially explained by the fact that the majority of ChIP-G4s correspond to canonical G4 motifs (Supplementary Fig. 2A), while >60% of oG4s contained long loops or noncanonical patterns[50]. It may also indicate that promoter G4s have

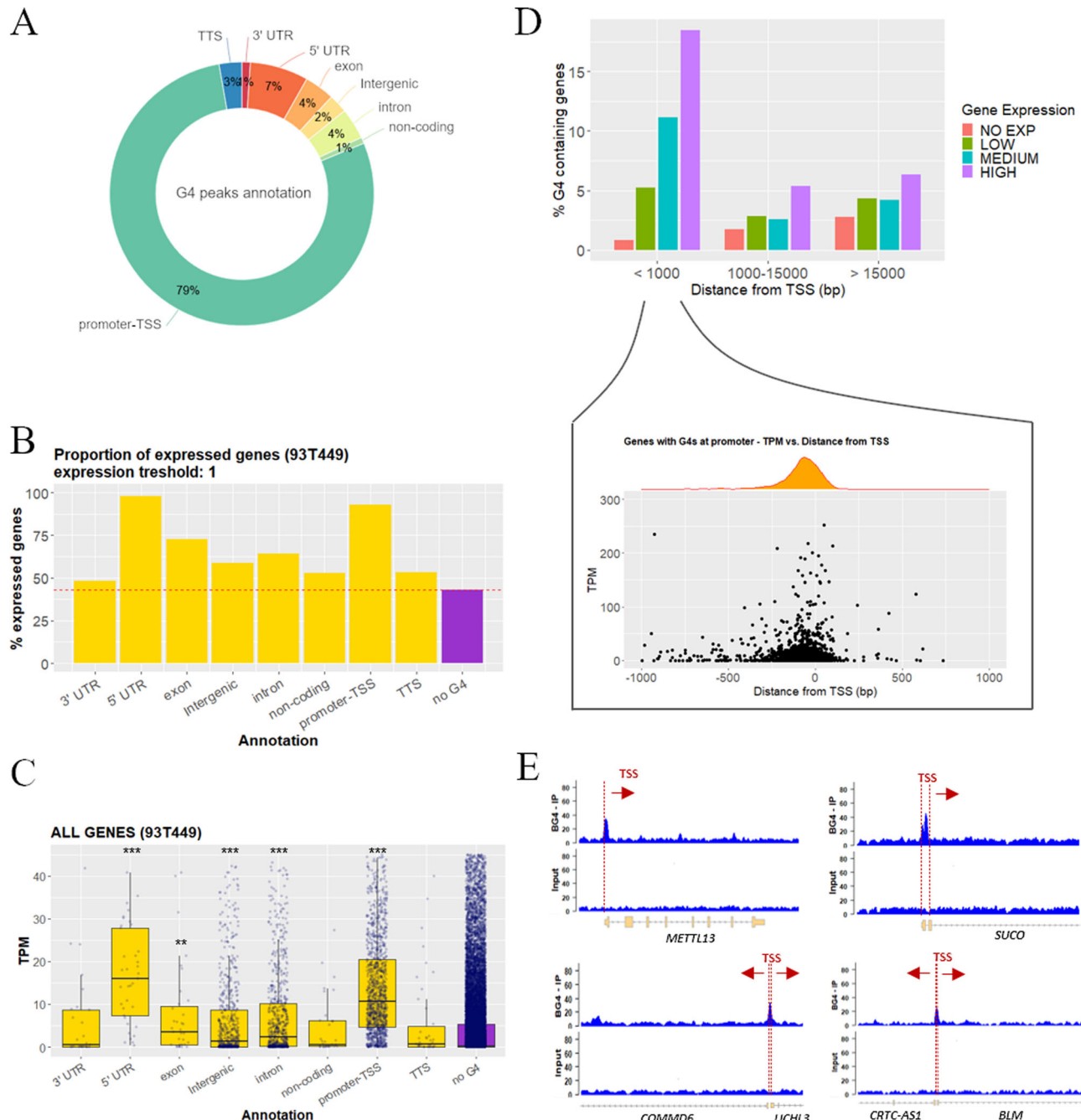

**Fig. 1 Genomic position of G4s and association to gene expression. A** Percentage distribution of G4 peaks in functional genomic regions according to Homer gene annotation. Percentages are normalized over the genomic abundance of each functional region. **B** Percentage proportion of expressed genes among the G4-containing genes (yellow). G4-depleted genes (no G4, violet) are reported as reference. One transcript per gene was considered as threshold. **C** Gene expression distribution expressed in transcripts per million (TPM) of all the G4-containing genes (yellow). Genes were grouped according to the functional annotation of the immunoprecipitated G4 region. G4-depleted genes (no G4, violet) are reported as reference. The box plots central line represents the median, the bottom, and upper bounds of the box represent the 25th and 75th percentile, respectively, and the whiskers represent the lowest and highest score, excluding outliers. The significance level of each gene category was calculated by two-sided *T* test (CI 95%) with respect to the no G4 group. \*\*\**p* value < 0.001, \*\**p* value < 0.01, the absence of asterisks indicates that the difference is not statistically significant. Exact *p* values are the following: 5′UTR *p* = 3.3e−14, exon *p* = 0.0032, intergenic *p* = 1.2e−9, intron *p* < 2.22e−16, promoter-TSS *p* < 2.22e−16. Numerosity of each category is: 3′UTR *n* = 29, 5′UTR *n* = 49, exon *n* = 44, intergenic *n* = 729, intron *n* = 808, noncoding *n* = 36, promoter-TSS *n* = 1434, TTS *n* = 47, no G4 *n* = 23662. **D** Upper panel: percentage of G4-containing genes, in genes grouped according to their expression level (no expression, low, medium, or high) and their distance from the TSS of the closest gene (<1000 bp, between 1000–15,000 and >15,000 bp). Lower panel: detailed view of gene expression level (TPM) and density distribution of genes with folded G4s within 1000 bp from TSS in function of the G4 distance from the TSS. **E** Genomic view of representative regions showing the G4-ChIP peak position with respect to the TSS: G4-ChIP peaks in two gene promoters with noncoding upstream regions are displayed in the upper panels (*METTL13* and *SUCO*); G4-ChIP peaks embedded in the coding regions of two adjacent genes with opposite transcription direction are shown in the lower panels (*COMMD6* and *UCHL3*—left; *CRTC-AS1* and *BLM*—right). Source data for each panel are provided or referenced in the Source data file.

peculiar features that discriminate them from G4s occurring in other genomic regions or that prevent them to be efficiently detected by G4-seq.

When analyzing the expression level of genes with ChIP-G4s, oG4s, or pG4s in their promoters, we found a significant increase of transcripts ($p$ value < 0.001), with ChIP-G4s with respect to pG4s and oG4s (Supplementary Fig. 5A). Among the G4-containing genes, those with ChIP-G4s and oG4s at promoter were significantly more expressed than all G4-harboring genes ($p$ value < 0.001). The same trend was visible in ChIP-G4s that did not overlap with oG4s, but it was lost in ChIP-G4s that overlapped with oG4s (Supplementary Fig. 5B). The latter result may indicate that G4-ChIP-seq and G4-seq preferentially recognize different structures. Since ChIP-G4s are captured in more physiological conditions, they are likely more representative. These data indicate that only the sequences that are actually folded into G4 are associated with high transcript levels, while pG4s or unfolded G4-forming sequences are not.

Analysis of accessible chromatin regions by Omni-ATAC-seq[51] showed that 93% of promoter ChIP-G4s (Fig. 2B) and 70% of all ChIP-G4s (Supplementary Fig. 6A) overlapped with open chromatin (Fig. 2A and Supplementary Data 3); 83% of open chromatin regions were embedded in gene promoters (Supplementary Fig. 6B); however, only 12% of these overlapped with G4s (Fig. 2B), suggesting that the G4 regulatory role is limited to specific DNA regions.

In 93T449 cells, the higher amount of actively transcribed genes fell into the category of genes embedded in open chromatin with folded G4s at their promoter: 90% of those genes were actively expressed. Lower abundances of actively transcribed genes were also found in G4-lacking promoters, but embedded in permissive chromatin (~80% of those genes) and in genes with G4-lacking promoters in silenced chromatin (~25%; Fig. 2C). When analyzing transcript abundance, we observed that the G4-folded promoters displayed higher transcriptional activity in open chromatin (Fig. 2D). The same trend was maintained in the other functional regions known to promote transcription (Supplementary Fig. 6C, D). These results suggest that G4s stimulate recruitment of the transcription machinery/chromatin remodeling factors; alternatively, G4 folding could be a control mechanism to restrain/reinitiate transcription in highly active genes.

To confirm the positive association of G4s with active transcription in WDLPS cells, we measured G4s foci by immunofluorescence in cells treated either with the HDAC inhibitor entinostat[52], stabilizing transcriptionally active chromatin, or the RNA-pol II inhibitor actinomycin D, preventing transcriptional elongation of the RNA chain[53] (Fig. 3). We observed high and dose-dependent increase in G4 foci in 93T449 cells upon entinostat treatment compared to the non-treated control (Fig. 3A, B, upper panel). This result confirms previous data obtained by ChIP-seq on keratinocytes[19] and indicates that folded G4 structures, and not the corresponding G-rich sequence, are associated with transcriptionally permissive chromatin regions. On the other hand, folded G4 foci in 93T449 cells decreased upon actinomycin D treatment (Fig. 3A, B, lower panel), further supporting the connection between transcription and G4 formation.

Specific gene programs are activated in cells to maintain their diversity and identity. Chromatin organization and gene expression are modulated by a complex interplay between epigenetic marks (DNA methylation, histone modification, and nucleosome positioning) and binding of core TFs to regulatory DNAs[54–56]. We hypothesized that G4s cooperated with TFs in the maintenance of cell-specific gene expression programs, contributing to the establishment of the transcriptome. To verify this issue, we compared transcriptome, chromatin state,

and ChIP-G4s in 93T449 cells vs HaCaT cells (Supplementary Data 4–6)[19]. The number of ChIP-G4s detected in the two cell lines was highly different (~5000 in 93T449 vs ~30,000 in HaCaT). Notwithstanding the substantially lower amount of 93T449 G4s, the identity and distribution of the majority of them was cell specific, i.e., not shared with HaCaT cells (Fig. 4D), and independent of chromosome length and number of encoded genes (Supplementary Fig. 7A).

The two cell lines showed a largely different expression pattern[19,57]: in 93T449 cells, 5541 and 3560 genes were expressed to a significantly higher and lower extent, respectively, compared to the same genes in HaCaT cells (Supplementary Data 7; $s$-value < 0.1; fold change > 1.0). Importantly, promoters contained folded G4s only in genes expressed at higher rates, while they were mainly unfolded when the same genes were downregulated (Fig. 4B). Despite associated to lower transcriptional output with respect to genes with promoter G4s, when G4s folded outside the promoter of the respective gene, we observed the same trend of differential expression when comparing the two cell lines (Supplementary Fig. 7C). This suggests that G4s sustain transcription also when folded outside promoters, by facilitating polymerase progression, double helix opening in gene coding sequence, and enhancer/promoter interaction in intergenic regions. Together, these data led us to the hypothesis that different cell types modulate G4 folding to establish or maintain their transcriptional program.

To verify if the G4 landscape defined the two cell lines, we performed pathway analysis on the G4-containing genes that were unique to the two cell lines (Supplementary Fig. 8A). The most represented pathways identified for the two cell lines were strikingly different: immune system-related pathways were prevalent in 93T449 cells, while vesicle-mediated, membrane trafficking, hemostasis, and cell growth signals prevailed in HaCaT cells. Interestingly, when comparing HaCaT cells to normal human keratinocytes (nHEK), we found 20 G4 peaks in key genes encoding for telomerase and components of its regulatory machinery (i.e., *TERC, TERT, TERF1, TERF2, TEP1, ACD, RTEL1*, and *POT1*), none of which were found to fold into G4s in nHEK[58]. These data perfectly fit with the notion that telomerase activity is one of the main causes of HaCaT cells immortalization, together with p53 mutations and the loss of senescence genes[59]. Using differential gene expression of G4-controlled genes as a parameter to identify cell-specific pathways, HaCaT cells displayed enrichment in genes involved in fibroblast growth factor receptors, typical of self-sufficiency in growth factor; cytochrome P450 related genes, which are a group of enzymes involved in skin protection[60–62]; DNA strand elongation and VEGFR vascular permeability and telomere synthesis, typically considered cancer hallmarks[63]. In contrast, 93T449 cells displayed enriched G4-driven pathways involved in immune system, suggesting a key role of immune system evasion, specifically mediated by Toll-like receptors 7/8 and 9, in this type of tumor[64,65]. We also found enrichment of O-linked glycosylation pathway and organelle biogenesis and maintenance, the activation of which is necessary for adipocyte differentiation and dynamic renewal[66,67]. Once again, these results support that G4s and gene expression are strictly connected and associated to stimulation/establishment of cell type and tumor-associated molecular pathways.

A recent work by Hänsel-Hertsch and colleagues[22] supported by previous literature[50,68–70] showed that cancer-related copy number variants (CNVs), i.e., amplifications, as well as single nucleotide variants, are enriched in breast cancer-specific G4s. To test whether WDLPS-specific G4s were associated with CNVs and also pan-cancer somatic variants (SVs), we calculated the overlap between 93T449 cell-specific CNVs or SVs[37], and G4 peaks found

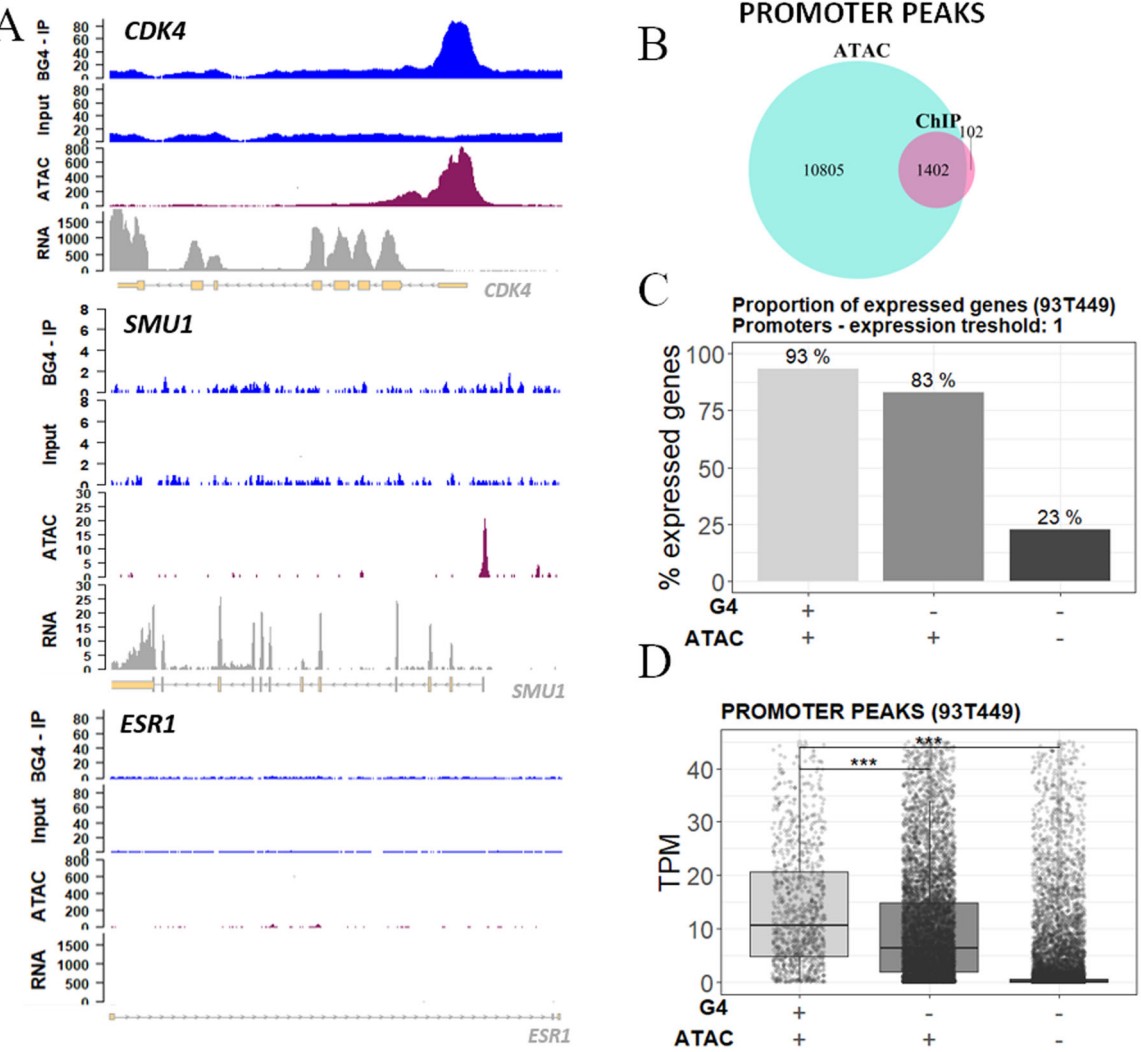

**Fig. 2 Relationship between G4s and open chromatin. A** Genomic view showing input and G4 IP samples (blue tracks), ATAC-seq (purple track), and RNA-seq (gray track) peaks in the promoter of representative genes: *CDK4* (upper panel), *SMU1* (mid panel), and *ESR1* as negative control gene (lower panel). **B** Venn diagram displaying the intersection between peak regions corresponding to IP G4s (light blue) and open chromatin regions (violet) mapped by ATAC-seq in promoters. **C** Percentage proportion of expressed genes grouped according to the presence of G4s and open chromatin signal in their promoter region. One transcript per gene was considered as expression threshold. **D** Expression distribution of all genes grouped according to the presence of ChIP-seq G4s and ATAC-seq signals in their promoter region. Gene expression is reported as TPM (transcript per million). In **C** and **D**, the presence and absence of G4 and ATAC-seq signals are indicated below the graphs. The box plots central line represents the median, the bottom, and upper bounds of the box represent the 25th and 75th percentile, respectively, and the whiskers represent the lowest and highest score, excluding outliers. The significance level of each category was calculated by two-sided *T* test (CI 95%) with respect to the G4:ATAC −/− group. ***p value < 0.001. Exact p values are <2.22e−16 for both G4:ATAC +/+ and −/+ categories. Numerosity of each category is the following: G4:ATAC +/+ n = 1351, −/+ n = 8893, −/− n = 16204. Source data for each panel are provided or referenced in the Source data file.

in 93T449 or HaCaT cells. We found that 73% of 93T449 CNVs (299 out of 411 detected CNVs) overlapped with folded G4s peaks, and 47% of all CNVs (193 out of 411) were in genomic regions that harbor 93T449 cell line-specific G4s (i.e., G4s that are not folded in the HaCaT cell line). On the contrary, only 1% of all CNVs colocalized with HaCaT-specific G4s (Supplementary Fig. 8C, D). This result supports previous observations indicating an association between G4s and CNVs[50]. In contrast, the abundance of different pan-cancer SVs (i.e., insertions, deletions, inversions, intra-, and inter-chromosomal translocations) in cell line-specific G4s was < 3%, thus indicating no correlation (Supplementary Fig. 8E). Comparing 93T449 genes affected by CNVs and harboring cell line-specific G4s with the same genes in HaCaT cells, we did not observe any significant difference in the general expression level (Supplementary Fig. 8F), suggesting that,

while G4s are associated to pan-cancer genomic instability regions, their role in gene expression regulation has a different mechanism, possibly recruitment of TFs.

Promoters (Fig. 4A) and other genetic regions (Supplementary Fig. 7B) associated with open chromatin were also different between the two cell lines. Analysis of the extent of different transcription levels in 93T449 vs HaCaT cells, evaluated according to the presence of folded G4s and open chromatin signals in promoters, indicated that the presence of both folded G4s and open chromatin strongly contributes to enhanced transcription levels (Fig. 4D, E, bars 2 and 3), to a higher extent with respect to regions with open chromatin only (Fig. 4D, bars 5 and 6, and Supplementary Data 5). Example regions for each of the described condition are reported in Supplementary Fig. 9. These data reinforce the indication that G4s are involved in the

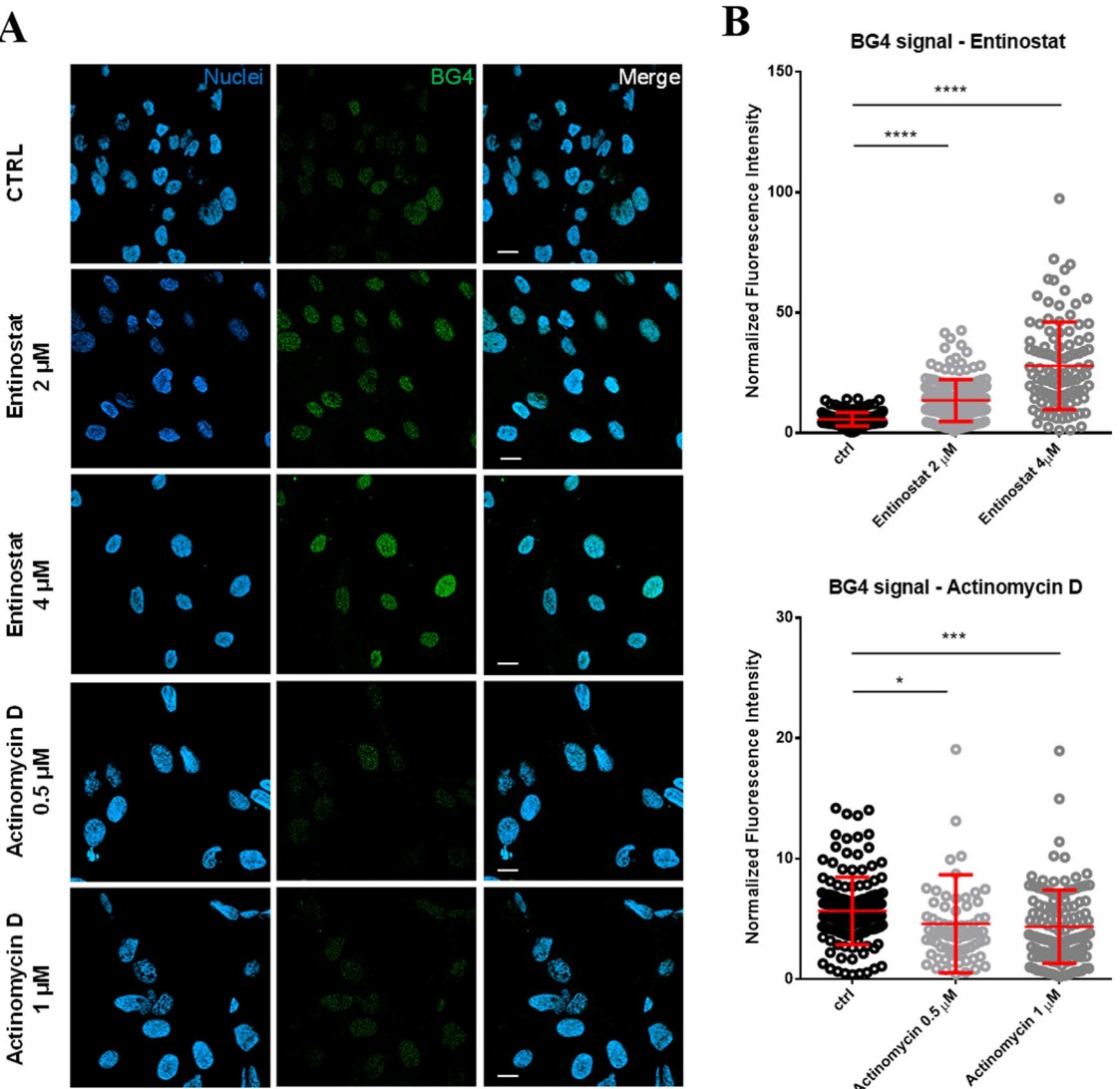

**Fig. 3 Detection of G4s foci upon transcription perturbation treatments. A** Representative fields of view showing G4 foci formation detected by immunofluorescence in control non-treated (CTRL), entinostat (2 and 4 µM) and actinomycin D (0.5 and 1 µM) treated 93T449 cells. Nuclear staining (blue), BG4 (green), and the merged channels are reported. Scale bars = 20 µm. The shown fields belong to one of two independent biological replicates. **B** Quantification of BG4 nuclear staining detected by immunofluorescence in control non-treated and entinostat (2 and 4 µM)—upper panel—or actinomycin D (0.5 and 1 µM) treated—lower panel. BG4 integrated fluorescence intensity within nuclei normalized by the corresponding nuclear area (µm$^2$) is reported. The central line for each condition represents the mean ± standard deviation. Statistical significance was calculated by unpaired two-sided $T$ test (CI 95%) with: ****$p$ value < 0.0001, ***$p$ value < 0.001, *$p$ value < 0.01. Exact $p$ values are the following: actinomycine D 0.5 µM $p = 0.0214$, actinomycine D 1 µM $p = 0.0002$, entinostat 2 µM $p < 0.0001$, entinostat 4 µM $p < 0.0001$. The number of quantified cells for each condition are the following: ctrl $n = 150$, entinostat 2 µM $n = 147$, entinostat 4 µM $n = 104$, actinomycin D 0.5 µM $n = 74$, actinomycin D 1 µM $n = 128$. Source data for each panel are provided or referenced in the Source data file.

recruitment of the transcriptional machinery or facilitate its function in open chromatin regions[22].

To test the above hypothesis that G4s in promoters stimulate TF binding, we first predicted the presence of putative TFBSs within the ChIP-G4s in 93T449 cells using Homer software. The most relevant TFBS corresponded to AP-1 (24.08% in ChIP vs 3.36% in background sequences, $p$ value 1e−490) and SP1 (24.98% in ChIP vs 9.45% in background sequences, $p$ value 1e −172; Fig. 5A, and Supplementary Data 8.1.1 and 8.2.1). AP-1 TFBSs were mostly centered at the G4 peak, while SP1 TFBSs were more broadly distributed within the peak, suggesting that they can either overlap or be adjacent to the G4 (Fig. 5B). When TFBSs were calculated on an extended promoter region (−1000 to +750 bp from TSS) of either ChIP-G4-enriched or G4-lacking

genes, different TFBSs were found and with consistently lower representation (Supplementary Fig. 10A). When considering all the target genes of AP-1 and SP1 (data from ENCODE database), those presenting G4s (ChIP-G4s, pG4s, and oG4s) were highly enriched with respect to those lacking G4s; among G4-containing categories, ChIP-G4s were the most enriched, suggesting the association between TF-binding and G4-folding capacity (Fig. 5C). This result also indicates that is not the simple GC richness or G4-folding potential to determine the binding of AP-1 and SP1, but the presence of folded G4 structures is a preferential condition for having TF binding. Another supporting observation that the correlation between TFs binding and G4 presence is not an artifact due to the GC richness of such regions, comes from the non-prominent GC frequency of AP-1 consensus sequence

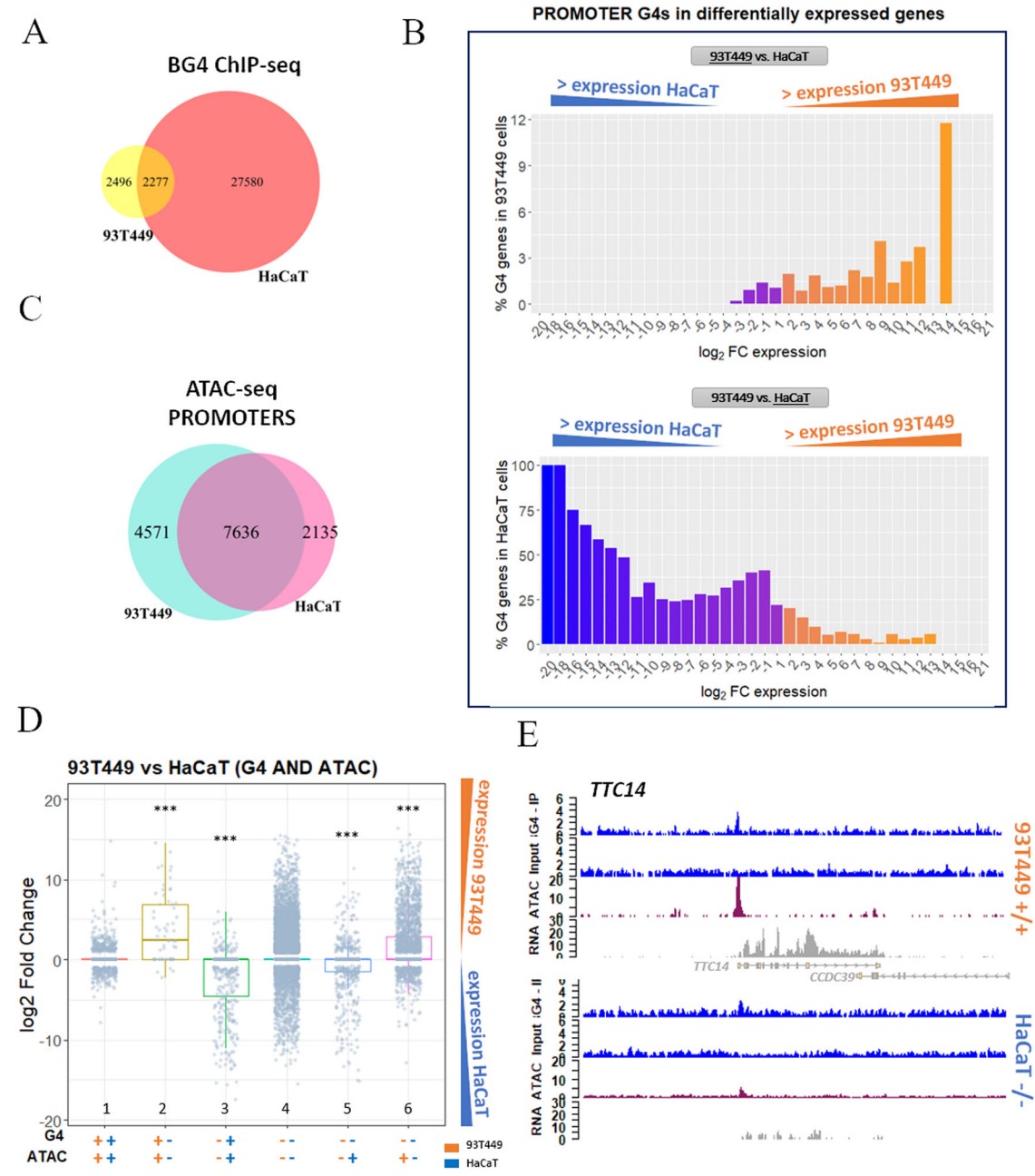

**Fig. 4 Comparison of 93T449 and HaCaT cell lines. A** Venn diagram showing the intersection between G4 peaks found in 93T449 (yellow) and HaCaT (salmon) cell lines. **B** Percentage of genes containing at least one G4 in their promoter in 93T449 (upper panel) or HaCaT (lower panel) cells, in function of their differential expression in the two cell lines expressed as $\log_2$ fold change (FC). Orange and blue-violet bars in both panels correspond to genes that have higher expression in 93T449 cells and HaCaT cells, respectively. **C** Venn diagram showing the intersection between the ATAC-seq peaks found in promoters of 93T449 (light blue) and HaCaT (pink) cell lines. **D** Differential gene expression comparison of the same genes in 93T449 and HaCaT cells, based on the presence of G4s and open chromatin combinations. Orange and blue symbols indicate data for 93T449 and HaCaT cells, respectively. The presence (+) or absence (−) of G4s or ATAC signals are reported. Bars indicate gene expression distribution of the differentially expressed genes in 93T449 vs HaCaT cells evaluated by two-sided *T* test in comparison to the G4:ATAC ++/++ condition (CI 95%, ***$p$ value < 0.001). Exact $p$ values are the following: +−/+− $p = 2.3e{-}16$, −+/−+ $p < 2.22e{-}16$, −−/−+ $p = 3e{-}8$, −−/+− $p = 4e{-}7$. The box plots central line represents the median, the bottom, and upper bounds of the box represent the 25th and 75th percentile, respectively, and the whiskers represent the lowest and highest score, excluding outliers. G4:ATAC ++/++ $n = 1098$, +−/+− $n = 61$, −+/−+ $n = 435$, −−/−− $n = 14{,}717$, −−/−+ $n = 649$, −−/+− $n = 2315$. **E** Genomic view showing representative regions of 93T449 and HaCaT cell lines gene expression (RNA-seq track, gray) with respect to the presence of G4 peaks (ChIP-seq, blue) and open chromatin (ATAC-seq, purple). In particular, *TTC14* gene is displayed, which shows both G4 and ATAC signals in 93T449 cells, while it shows limited accessibility and no G4 in HaCaT cells. These differences are reflected in the corresponding RNA amount, which is much lower in HaCaT cells. Source data for each panel are provided or referenced in the Source data file.

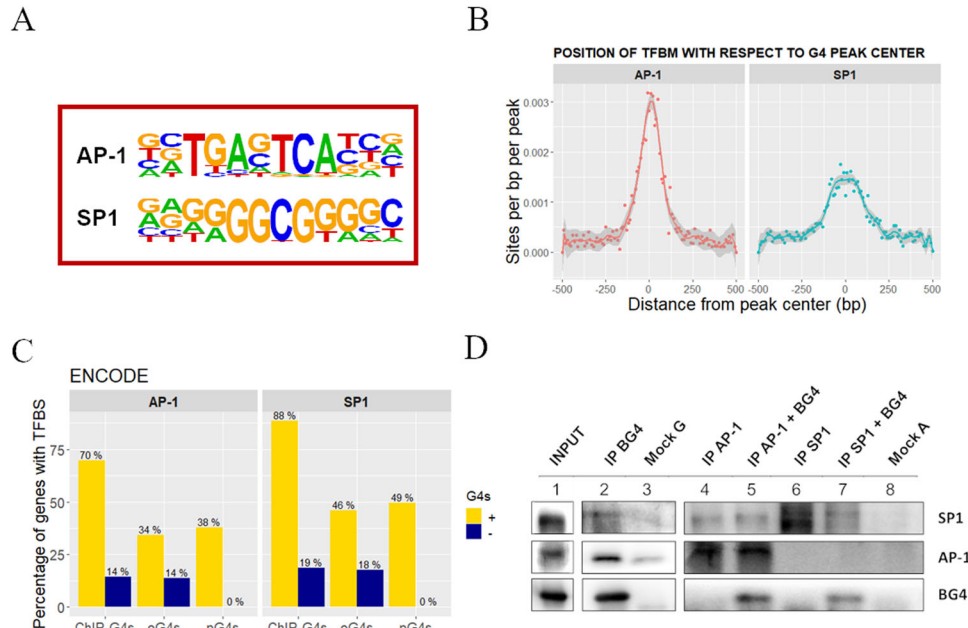

**Fig. 5 Identification of TFs binding to BG4-IP regions in 93T449 cells. A** Consensus sequences of TFBSs that are significantly enriched in BG4 ChIP peaks, as calculated by Homer software. AP-1: *p* value 1e−490, SP1: *p* value 1e−172. **B** Position and frequency of TFBSs with respect to the BG4 ChIP peak center. Data points for each TFBS occurrence are reported and fitted according to a nonparametric spline regression curve, the gray area surrounding the curve represents the confidence interval as measure of the regression likelihood. **C** Percentage of genes with (yellow) and without (darkblue) ChIP-G4s, oG4s, and pG4s containing validated TFBS for AP-1 and SP1, according to ENCODE database. **D** Western blot showing co-immunoprecipitation of G4s and the two TFs SP1 and AP-1. The INPUT lane 1 corresponds to the total fraction of the sheared chromatin used as starting material. G4s were immunoprecipitated by BG4 antibody (IP BG4), and AP-1 and SP1 were detected by immunoblotting (lane 2). AP-1 (lanes 4 and 5) and SP1 (lanes 6 and 7) TFs were immunoprecipitated from the sheared chromatin with or without previous incubation in the presence of BG4 antibody. Mock G (lane 3) and A (lane 8) are the negative controls immunoprecipitated in the absence of primary antibody using protein-G- or protein-A-coated beads, respectively. The shown blots belong to one of at least two independent biological replicates performed for each sample. Source data for each panel are provided or referenced in the Source data file.

(ATGAGTCA). Moreover, AP-1-binding site is strongly over-represented at the peak center of ChIP-G4s (Fig. 5B), while the highest GC frequency within the G4 peak, is observed at ~50 bp far from the peak center, (Supplementary Fig. 1D) supporting the absence of correlation between intrinsic GC richness and AP-1-binding site. Less obvious is the case of SP1, due to the intrinsic GC richness of its consensus sequence (GGGGCGGGG), which could participate itself to G4 folding.

As a further proof of their capacity to interact with regions of folded G4s, SP1 and AP-1 were co-immunoprecipitated using anti-G4 primary antibody (Fig. 5D, lanes 2 and 3) and, vice versa, primary antibodies against SP1 and AP-1 co-immunoprecipitated with the anti-G4 antibody added to the samples as marker of G4 regions (Fig. 5D, lanes 4–8), proving that these TFs interact with their binding sites in the presence of folded G4s (Fig. 5D). These data corroborate the observation that G4s and TFs SP1 and AP-1 are strictly linked to transcriptionally active regions. The co-occurrence of G4s and TFs could be (i) independent and due to G4-stimulating conditions typical of highly transcriptionally active DNA regions (such as negative supercoiling) or (ii) they can have a cooperative role in which G4s are exploited by cells to facilitate TF interaction with DNA. The latter effect can be reached by G4-mediated exposure of the binding region or altered DNA methylation state, as G4s were reported to prevent CpG island methylation, a condition which is required for transcription initiation[21,56,71]. An alternative possibility is that TF binding to G4 regions stimulates G4 folding to maintain the underneath chromatin in a transcription permissive state.

SP1 and AP-1 are two master TFs that regulate many fundamental processes, such as cell differentiation, growth, apoptosis, immune and DNA damage response, and chromatin remodeling[72,73]. AP-1 TF is a dimeric complex composed of proteins belonging to the JUN (JUN, JUNB, and JUNC), FOS (FOS, FOSB, FRA1, and FRA2), ATF (ATF1-4, ATF-6, BATF, and ATFx), and MAF family (c-MAF, MAFA, MAFB, MAFG/F/K, and NRL). AP-1 activity varies according to the cell type and it is modulated by its dimer composition, which in turn is determined by the differential expression of its components and the sequence of the available AP-1-binding sites[73,74]. SP1 activity is regulated by post-translational modifications and, when over-expressed, it is generally considered a negative prognostic factor for cancer[75]. When we analyzed AP-1 and SP1 expression in 93T449 and HaCaT cells, we observed different expression of several of the AP-1 TF components between the two cell lines, while SP1 did not show any significant variation (Supplementary Fig. 10B). This observation supports the connection between G4s and TFs, since the different composition of AP-1 dimers reflects its different activity and affinity for binding sites and possibly also the modulation of different cell line-specific G4s.

**Discussion**

In the present work, we tested whether the dynamic folding of G4s in vivo contributed to define the cell-specific chromatin organization in different cell lines. Being the transcriptional landscape the major cell signature, we hypothesized that, if G4s are involved in cell type definition, they would contribute to establish the transcriptome or modulate chromatin accessibility. Indeed, WDLPS G4s are highly prevalent at promoters and the corresponding genes are more prone to be transcriptionally active and they lead to higher transcript levels. G4 proximity to the gene

TSS and an open chromatin context enhance this effect. Even though G4s sterically impair RNA polymerase II processivity in vitro[7–13], in cells G4 formation occurs during the partial opening of the DNA double helix[76], and thus G4 association with highly transcriptionally active genes, highlighted in the present work and other recent studies[19,30], has rational bases.

Our results obtained in a mesenchymal cell line were compared with data obtained in endothelial cells[19]. The two cell lines 93T449 and HaCaT, with their inherently different transcriptional and epigenetic states, were chosen to better appreciate cell type-specific differences in G4 landscape, information that was not obvious from the current literature[21,56]. We found that the same G4-forming sequence is either folded or unfolded in the two cell types based on the transcription state of the corresponding gene. Pathway enrichment analysis of differentially expressed G4-regulated genes highlighted G4 involvement in cancer- or cell type-related biological processes. For 93T449 cells, these data provide precious information, given the limited knowledge of the exact mechanisms leading to tumorigenesis and the lack of information on normal adipocytes. G4 enrichment in cell-specific CNV sites, but not SVs, further supports the already known involvement of G4s in genome instability[70]; in this case, however, the transcriptional output was not affected, suggesting that G4-mediated regulation of transcript levels occurs prevalently by mechanisms other than alteration of gene copy number. Perturbation of transcription by chemical compounds stabilizing permissive chromatin (entinostat) or inhibiting RNA polymerase II elongation (actinomycin D) led to dose-dependent increase or decrease of G4s foci, respectively. Besides strengthening the relationship between G4s folding and gene transcription, this result provides useful insights in the comprehension of G4 transcriptional regulatory role. In this regard, with the state of the art data, two hypothesis may hold: (i) G4s folding is a feedback mechanism induced after elevated transcription to contain or repress the process; and (ii) folded G4s enhance transcription by favouring binding of TFs and chromatin remodeling agents, or holding the DNA double helix open, facilitating reinitiation of transcription[29,77,78]. Our present data make us lean towards the second hypothesis, since we found significant enrichment of AP-1- and SP1-binding sites in regions overlapping or closely flanking the folded G4s. Co-immunoprecipitation of G4s and AP-1 or SP1 demonstrated that the G4-folded state is permissive to TF binding. Moreover, we highlighted that the correlation between TFs binding and G4s depends on the presence of folded G4 structure, rather than the intrinsic GC richness of the G4-forming sequences. These data strengthen the role of G4s as chromatin marks and suggest that the effect of G4s on transcription can also involve the specific recruitment of cell-specific TFs. AP-1 is composed of a dimer of JUN and FOS family proteins, and depending on the signal and context it can switch on or off different transcriptional programs. It has been shown that AP-1 induces chromatin structure changes[79,80] and DNA bending, which in turn increases site affinity for AP-1 (refs. [81,82]). It is possible that G4 formation and the distortion generated in the nearby DNA portions act similarly, in the end facilitating AP-1 binding. In addition, AP-1 binding has bene reported to be altered by CpG island methylation[83]. Since G4s have been shown to protect CpGs from methylation, we speculate that G4 formation in chromatin is a favorable mark for the recruitment of AP-1. We showed that expression of AP-1 components is different in 93T449 and HaCaT cells, suggesting that the different composition of the AP-1 complex is also associated with the cell type-specific G4 landscape. Similarly, SP1 activates transcription by triggering formation of the transcription pre-initiation complex or driving chromatin toward the permissive state[84]. SP1 is also involved in the maintenance of the methylation-free state of

CpGs, required for the activation of gene expression. Despite being ubiquitous, SP1 activity is tissue- and development-specific giving rise to differential expression of cell-specific genes. In this context, one determining factor are epigenetic changes modifying the SP1-binding site availability[85]. Our data hint at G4s being the epigenetic marks responsible, at least in part, for the regulation of SP1 recruitment.

Our work deepens the knowledge on how gene expression landscape of specific cell types is established and maintained. It proposes a model where G4s cooperate with TFs to shape the cell transcriptome. It also indicates possible new therapeutic targets, such as *MDM2* and *CDK4* promoter G4s, for a disease as challenging as WDLPS.

## Methods

**Primers and oligonucleotides**. Desalted primers and oligonucleotides were purchased from Eurofins (Munich, Germany) or Sigma Aldrich (Milan, Italy). A detailed list of primers name and sequence is available in the Supplementary Table 2.

**BG4 expression and purification**. BG4-encoding plasmid (kindly provided by Professor Shankar Balasubramanian, University of Cambridge, UK) was transformed into BL21(DE3) competent cells (Stratagene), which were cultured in TY medium (1.6% tryptone peptone, 1% yeast extract, and 0.5% NaCl) and 50 μg/ml kanamycin. Transformed cells were grown at 37 °C 160 r.p.m. to an OD600 of 0.7–0.8. BG4 antibody expression was induced with 0.85 mM isopropyl β-D-1-thiogalactopyranoside overnight at RT. The cells were pelleted for 25 min at 25,000 × g at 4 °C, resuspended in lysis buffer (20 mM Tris-Cl pH 8.0, 50 mM NaCl, 5% glycerol, 1% Triton, and 100 μM phenylmethanesulfonylfluoride solution), and lysed through five cycles of freezing and thawing. After centrifugation at 10,000 × g at 4 °C for 20 min, the supernatant was filtered (0.45 μm) and purified on a Protino Ni-NTA-Agarose Affinity column (Machery-Nagel, Germany), according to the manufacturer instructions. The column was washed in 20 mM imidazole in 20 mM Tris-HCl pH 8.0 and 300 mM NaCl, and BG4 antibody 1.5 ml fractions eluted in 250 mM imidazole in 20 mM Tris-HCl pH 8.0 and 300 mM NaCl. BG4 antibody containing fractions were checked on a Coomassie-stained SDS–PAGE and concentrated in Amicon Ultra-3k Centrifugal Filter Unit (Millipore). The concentration of BG4 was determined using Thermo Scientific Pierce BCA Protein Assay kit, and the antibody was stored at −20 °C.

**Electrophoretic mobility shift assay**. EMSA was used to validate BG4-specific binding to G4 structures. The oligonucleotides for *c-myc*, *bcl-2*, *c-kit* G4s, and a non G4 single-stranded G-rich sequence named *scrambled* (Supplementary Table 2) were 5′-end labeled with [γ-32P]ATP by T4 polynucleotide kinase after 30 min incubation at 37 °C. After DNA precipitation, labeled species were resuspended in lithium cacodylate buffer (10 mM, pH 7.4, KCl 100 mM), heat denatured, and folded at room temperature.

Binding reactions were performed in binding buffer (Tris-HCl 50 mM pH 8.0, KCl 100 mM, NaCl 50 mM, glycerol 8%, and protease inhibitor cocktail—Sigma Aldrich, Milan, Italy), with 40 nM labeled oligonucleotides and growing concentrations of purified BG4: 0, 1.5, 3.5, and 5.0 μg. The reactions were incubated for 1 h 30 min at 37 °C. Samples were loaded on 8% non-denaturing polyacrylamide gels for ~15 h at 40 V. Gels were dried using a gel dryer (Bio Rad Laboratories, Milan, Italy) and visualized by phosphorimaging (Typhoon FLA 9000, GE Healthcare Europe, Milan, Italy).

**G4 chromatin immunoprecipitation**. The 93T449 (ATCC® CRL-3043™) cells were grown to 80% confluence in RPMI 1640 (Gibco, ThermoFisher Scientific, Waltham, MA, USA) supplemented with 10% heat-inactivated FBS. After trypsinization, 2 million cells were fixed in RPMI containing 1% (v/v) formaldehyde and 10% (v/v) FBS for 10 min at RT. After 5 min quenching with 125 mM glycine, cells were pelleted and washed twice with PBS containing 10% FBS. The flash-frozen pellets were lysed for 5 min on ice in 100 μl of 50 mM Tris-HCl pH 8.0, 10 mM EDTA, 0.5% SDS, and protease inhibitor cocktail. Samples were sonicated using the Covaris E220 to shear chromatin to an average size of 100–500 bp (2% duty cycle, 105 W peak incident power, 200 cycles per burst, 25 min). Sheared chromatin was diluted 1:5 in IP-buffer (10 mM Tris-HCl pH 7.5, 1 mM EDTA, 0.5 mM EGTA, 1% Triton X-100, 0.1% SDS, 0.1% Na-deoxycholate, and 140 mM NaCl) supplemented with protease inhibitor cocktail. After centrifuging 10 min at 13,000 × g at 4 °C, the supernatant containing soluble chromatin fraction was recovered and incubated with 0.7 mg/ml RNase A (ThermoFisher) for 30 min at 37 °C. For chromatin immunoprecipitation, 10 μl protein-G magnetic beads (Pierce™ ThermoFisher) were washed in IP-buffer and incubated with 1 μg Anti-FLAG Ab (Sigma Aldrich #F3165) for 1 h at 4 °C on a rotating wheel. A total of 50 μl of RNA digested chromatin were incubated with 250 ng BG4 Ab (or without for the Mock negative control) for 1 h at 16 °C. The anti-FLAG-coated beads were washed with IP-buffer

and incubated with chromatin–BG4 complex for 3 h at 4 °C on a rotating wheel. Beads were washed four times with IP-buffer and once in wash buffer (10 mM Tris-HCl pH 8.0 and 10 mM EDTA). Elution of immunoprecipitates and chromatin crosslink reversal were performed incubating beads with 70 µl elution buffer (10 mM Tris-HCl pH 8.0, 5 mM EDTA, 300 mM NaCl, and 0.5% SDS) containing 0.3 mg/ml RNase A (ThermoFisher) for 30 min at 37 °C followed by the addition of 0.5 mg/ml proteinase K for 1 h at 55 °C and 8 h at 65 °C shaking. Supernatant was then recovered and incubated for one additional hour at 55 °C in the presence of 0.25 mg/ml proteinase K (ThermoFisher). The eluate was finally purified with SPRI AMPure XP beads (Backman Coulter). For each technical replicate, eluted DNA from two ChIP reactions were combined and the pool subjected either to G4 enrichment quantification via qPCR or to to library preparation for sequencing.

**G-quadruplex ChIP-qPCR.** The immunoprecipitated sample (IP and Mock) and the input were used to quantify G4 enrichment via qPCR, using Fast SYBR PCR mix (Applied Biosystems), with a LightCycler 480 (Roche) quantitative PCR machine. Cycling conditions were 95 °C for 20 s followed by 50 cycles of 3 s at 95 °C and 30 s at 60 °C. We employed primer pairs that target G4-ChIP-positive and -negative regions (Supplementary Table 2). Relative enrichments were derived with respect to their inputs.

**G4-ChIP-seq library preparation.** The immunoprecipitated sample and the input were subjected to Nextera library preparation using NEBNext Ultra II DNA library Prep Kit for Illumina(NEB). The quality and size of libraries and chromatin shearing fragments were checked by Agilent Bioanalyzer, using Agilent DNA High Sensitivity Chips (Agilent Technologies). Samples were sequenced on an Illumina HiSeq 1500 platform in single-end using 50-bp reads. The experiment was repeated twice.

**Putative G4 prediction.** The presence of pG4s in the BG4 immunoprecipitated peaks was assessed by two different computational tools: (i) Quadparser, based on a regular expression matching algorithm[3]; and (ii) G4Hunter, based on a scored sliding window approach that considers also the G richness and skewness of all bases in each window[45,86]. A fasta file containing the sequences of BG4 immunoprecipitated regions was obtained from Homer (http://homer.ucsd.edu/homer/) output by mean of bedtools[87] and used as input of both prediction algorithms. Quadparser script was downloaded from https://github.com/dariober/ as indicated by Puig Lombardi et al.[86], and applied with two different regular expressions, in order to allow the matching of loops with length 0–7 ([gG]{2,5}\w{0,7}){3,}[gG]{2,5} or 0–12 ([gG]{2,5}\w{0,12}){3,}[gG]{2,5}. G4Hunter algorithm was instead retrieved from https://github.com/AnimaTardeb/G4Hunter and applied using a window size of 15 bp and score threshold of 1.25, demonstrated to reliably discriminate G4s from non-G4s sequences[86]. The obtained results were then evaluated by using R programming language.

**RNA extraction and cDNA library preparation.** Total RNA for RNA-seq experiments was extracted from 80% confluent cells. Then, 1 million cells were trypsinized, pelleted, and resuspended in 1 ml TRIreagent (Sigma Aldrich). Total RNA was extracted by phenolchloroform and purified through the RNA Clean and Concentrator-25 kit (Zymo Research), following the manufacturer's instructions. Ribosomal RNA was depleted and purified using Ribo-Zero rRNARemoval Kit (Illumina) and RNA Clean and Concentrator-5 (Zymo Research). The quality of extracted RNA was checked by Agilent Bioanalyzer on Agilent RNA 6000 Pico Chips (Agilent technologies) both before and after rRNA depletion. RNA-seq libraries were generated using the NEBNext Ultra Directional RNA Library Prep kit for Illumina (NEB). Agilent DNA High Sensitivity Chips (Agilent Technologies) were used to check library size and quality. Samples were sequenced on an Illumina HiSeq 1500 platform in single-end using 50-bp reads. The experiment was done in three independent biological replicates.

**Comparison of pG4s, oG4s, and ChIP-G4s.** To obtain direct evidence of the association between G4s in the folded state and transcriptional level, three G4 categories were compared, namely: pG4s, oG4s, and ChIP-G4s. This comparison was used to distinguish the transcriptional effect of a G4-permissive sequence vs the folded G4 structure. pG4s correspond to the Quadparser[3]-predicted G4s with G-tracts of 2–5 Gs and loops length 0–12 nts; oG4s are G4s forming sequences detected by modified Illumina sequencing protocol from Chambers et al.[50,88]; ChIP-G4s are the BG4 immunoprecipitated regions in 93T449 cells. Bed files containing the G4 regions of each category were annotated using Homer software (http://homer.ucsd.edu/homer/) to retrieve the corresponding gene, and then integrated with 93T449 RNA-seq data to evaluate their expression level.

**ATAC-seq (assay for transposase-accessible chromatin followed by high-throughput sequencing).** ATAC-seq of 93T449 cells was performed according to the Omni-ATAC protocol developed by Corces et al.[51]. Briefly, 50,000 viable cells were pelleted and lysed in resuspension buffer (Tris-HCl pH 7.4 10 mM, NaCl 10 mM, MgCl$_2$ 3 mM, NP40 0.1%, Tween-20 0.1%, and digitonin 0.01%—Promega #G9441) for 3 min on ice. Next, nuclei were pelleted and the transposition reaction

was performed incubating the lysate for 30 min at 37 °C under agitation in the presence of transposition mixture (Tris-HCl pH 7.6 10 mM, MgCl$_2$ 5 mM, dimethyl formamide 10%, Tn5 enzyme 100 nM—Illumina #20018704, digitonin 0.01% —Promega #G9441, Tween-20 0.1%, and PBS 33%). The transposition reaction was purified (Qiagen MinElute PCR Purification kit, Qiagen #28004) and pre-amplified by PCR in the presence of adapter primers (NEBNext 2× PCR Master Mix, New England Biolabs Inc. #M0541S). Amplification was monitored by RT-PCR using the PowerUp™ SYBR™ Green Master Mix (ThermoFisher Scientific, #A25741) to determine the number of additional amplification cycles[89]. Finally, samples were purified with SPRI AMPure XP beads (ThermoFisher Scientific) and sequenced on an Illumina HiSeq 1500 platform in single-end using 50-bp reads.

**G4 Immunofluorescence imaging and data quantification.** For immuno-fluorescence studies, 93T449 cells were seeded at $4 \times 10^4$ cells/well on glass 12-mm-diameter coverslips in 24-well plates and grown overnight at 37 °C. Following treatment with entinostat (EPS002, Sigma Aldrich) 2–4 µM or actinomycin D (A1410, Sigma Aldrich) 0.5–1 µM for 48 h, cells were fixed in 2% paraformaldehyde in PBS 1× for 20 min at room temperature. Staining with BG4 was performed as previously reported[14] with minor modifications: cells were permeabilized with 0.1% Triton X-100 (Sigma Aldrich) in PBS 1×, treated with 50 µg/ml RNase A (EN0531, ThermoFisher), blocked using BlockAid (B10710, ThermoFisher), incubated with BG4 1:100 (stock 0.25 mg/ml, MABE917, Merck), M2 anti-FLAG 1:800 (F3165, Sigma Aldrich), and anti-mouse Alexa 488 1:500 (A21204, ThermoFisher) antibodies. Nuclei were stained with TOTO3 (1:2000, ThermoFisher) for 20 min at RT. Digital images were acquired using a Nikon A1R Laser Scanning confocal microscope equipped with NIS-Elements Advanced Research software (Nikon Instruments), with 60× objectives. For BG4 (green channel), images were visualized at 488 nm excitation wavelength and 500–550 nm emission wavelength; for cell nuclei (blue channel), 641 nm excitation wavelength and 663–738 nm emission range were applied.

Confocal imaging quantification was performed using ImageJ software. 2D nuclear dye staining was employed for nuclei identification and nuclear area determination. The integrated fluorescence intensity values for BG4 channel were background subtracted and normalized to the nuclear area.

**Identification of G4-associated transcription factors binding sites.** Bed files of 93T449 ChIP-G4s, Quadparser-predicted pG4s (G 2–5, loops 0–12), and oG4s from Chamber et al.[50] were used to annotate peaks and extract genes with promoter G4s by mean of Homer software (http://homer.ucsd.edu/homer/). The complete list of human genes annotated on the GRCh38-hg38 reference genome was retrieved from BioMart Ensembl database (http://www.ensembl.org/biomart/martview) and used to divide genes into promoter G4-containing/-depleted genes, according to the three categories of ChIP-G4s (Supplementary Data 8.1 and 8.2 for known motifs, and Supplementary Data 8.1.1 and 8.2.1 for de novo motifs), pG4s, and oG4s. The findMotifs.pl function of Homer software was employed to predict the presence of TFBSs associated to either all the ChIP-G4s peaks and genes with/without promoter ChIP-G4s, pG4s, and oG4s. Since promoter ChIP-G4s were strongly clustered within −1000 and +750 bp form the corresponding gene TSS, we set this region for the prediction of TFBSs. To confirm G4 association with the predicted TFs, the validated target genes of AP-1 and SP1 were retrieved from ENCODE database, accessed through Harmonizome[90].

**Co-immunoprecipitation of G4s and transcription factors and western blotting.** Co-immunoprecipitations to detect G4s interaction with TFs were performed by immunoprecipitating either BG4–G4 complex, or AP-1 and SP1 TFs. In the first case, G4s immunoprecipitation was performed as described for the BG4 ChIP-qPCR procedure, but after the last wash, three replicates of BG4 captured material were collected and eluted in 35 µl elution buffer (10 mM Tris-HCl pH 8.0, 5 mM EDTA, 300 mM NaCl, and 0.5% SDS) by incubating beads for 1 h at 55 °C to revert the cross-linking without degrading proteins. The collected proteins are expected to be G4s interacting proteins, since they were co-immunoprecipitated together with BG4. The presence of BG4 (internal positive control) and the TFs AP-1 and SP1 was then checked by western blot, as described below.

For the immunoprecipitation of AP-1 and SP1, the fixed and sheared chromatin from 1.5 million cells was treated with 0.7 mg/ml RNase A (ThermoFisher) for 30 min at 37 °C, precleared with protein-A magnetic beads (Pierce™ ThermoFisher) for 30 min at 4 °C under rotation to reduce the background due to the nonspecific adhesion of sample to the beads. A total of 20 µl protein-A magnetic beads (Pierce™ ThermoFisher) were washed in IP-buffer and incubated with 4 µg anti-AP-1 (Thermo Scientific™ #MA5-15172) or anti-SP1 Ab (ChIPAb+™ Merck #17-601) for 1 h at 4 °C on a rotating wheel. After incubation in the presence or absence of 250 ng BG4 for 1 h at 16 °C, the precleared chromatin was incubated on the AP-1, SP1 functionalized beads, or nonfunctionalized protein-A beads as Mock for 4 h at 4 °C under rotation, washed four times with IP-buffer and once in wash buffer (10 mM Tris-HCl pH 8.0, and 10 mM EDTA). After the last wash, three replicates were collected and eluted in 35 µl elution buffer (10 mM Tris-HCl pH 8.0, 5 mM EDTA, 300 mM NaCl, and 0.5% SDS) by incubating beads for 1 h at 55 °C to revert the cross-linking without degrading proteins. If AP-1 and SP1 TFs interact with G4s, BG4 antibody is supposed to be co-immunoprecipitated. Thus, western

blotting was employed to detect AP-1, SP1 (internal positive controls), and BG4 in the immunoprecipitated samples.

For western blotting of both co-immunoprecipitation approaches, the eluted proteins were quantified by Thermo Scientific Pierce BCA Protein Assay kit—the INPUT, IP, and Mock (negative control immunoprecipitated without BG4) were next loaded on an SDS–PAGE denaturing gel. The gel separated proteins were transferred on a PVDF membrane, blocked in TBS-tween 0.1% buffer supplemented with 5% BSA, incubated with primary antibodies (anti-AP-1 1:1000 Thermo Scientific™ #MA5-15172, anti-SP1 1:1000 ChIPAb+™ Merck #17-601, anti-FLAG 1:1000 Sigma Aldrich #F3165), washed in TBS-tween 0.1%, next incubated with secondary goat anti-rabbit 1:4000 (Merck-Millipore #12-34), and goat anti-mouse 1:4000 (Merck-Millipore #12-349) HRP antibodies. Images were acquired on a Uvitec instrument by reading HRP bioluminescence.

**Data analysis**. Raw FASTQ reads were trimmed to remove adaptor contamination and aligned to the primary assembly of the human reference genome version GRCh38 using Bowtie1 (http://bowtie-bio.sourceforge.net/index.shtml). Reads with more than two mismatches and multimapped reads were excluded from further analysis. G4-ChIP peaks were identified and mapped using Homer (http://homer.ucsd.edu/homer/index.html). Only peaks with at least twofold more normalized tags count in the target experiment with respect to the input (used as control) were considered, the analysis was performed with disabled local tag count and Poisson $p$ value threshold of 0.0001. Homer was next used to associate peaks with the nearby gene, determine the genomic annotation of the region occupied by the peak and merge peaks from replicates. To assess the confidence of peak calling, the results obtained by Homer were compared to MACS2 (ref. [91]), applied with the same parameter described at URL https://github.com/sblab-bioinformatics/dna-secondary-struct-chrom-lands[19].

RNA-seq reads were aligned to the human reference genome with TopHat and filtered by using samtools[92] to remove alignments with quality <20, not primary alignments, and PCR duplicates. Gene expression levels were quantified as transcripts per million. Genes differentially expressed between 93T449 cells and HaCaT cells and ($s$-value < 0.1; fold change > 1.0) were identified using the Bioconductor package DESeq2 and "apeglm" for LFC shrinkage[57].

ATAC-seq peaks were identified and mapped using Homer and employing the ChIP-seq input sample as genomic reference. A fixed peak size was estimated by the software on the basis of autocorrelation analysis. The other parameters were set to default. Bedtools were used to merge Homer peak files into a single bed file[87]. Homer was next used to associate peaks with the nearby gene and determine the genomic annotation of the region occupied by the peak. For the intersection of ChIP-seq and ATAC-seq peaks, the bedtools intersect function was employed.

All the further statistical analysis was performed using R[93].

**Pathway enrichment, CNVs, and SVs analysis**. The R package "ChIPpeakAnno"[34,94] combined with Reactome database (https://reactome.org/) was employed to calculate significant pathway enrichment for genes that harbor G4s, according to BG4 ChIP-seq data in 93T449 and HaCaT cell lines. Differentially expressed genes filtered for the presence of G4s were also considered for pathway enrichment calculation. In both cases were considered only the Reactome database pathways with a minimum of ten terms in the human genome and $p$ value threshold for pathway enrichment significance was set at 0.01.

The R package "ChIPpeakAnno"[34,94] was employed to calculate overlapping of 93T449 CNVs and SVs regions with G4-ChIP-seq peaks. CNVs data obtained by BIC-seq and SVs obtained by WGS (Crest) were retrieved from Macchia et al.[37]. Genomic regions coordinates were converted from GRCh37/hg19 to GRCh38/hg38 by mean of UCSC Lift Genome Annotations tool (https://genome.ucsc.edu/cgi-bin/hgLiftOver).

**Reporting summary**. Further information on research design is available in the Nature Research Reporting Summary linked to this article.

## Data availability
All genomic data produced in the present project (93T449 G4-ChIP-seq, ATAC-seq, and RNA-seq) have been deposited in the NCBI GEO database under accession number GSE145543. HaCaT cells datasets for G4-ChIP-seq, ATAC-seq, and RNA-seq were instead downloaded from GEO at the following accession number GSE76688. The oG4s dataset produced in the presence of K+ was retrieved from GEO at the following accession number GSE63874. The list of CNVs and SVs for 93T449 cells obtained by BIC-seq and WGS, respectively, are available as online supplementary material to the paper ref. [37] (https://doi.org/10.1534/genetics.117.300552). The validated target genes of AP-1 (https://maayanlab.cloud/Harmonizome/gene_set/JUN/ENCODE+Transcription+Factor+Targets) and SP1 (https://maayanlab.cloud/Harmonizome/gene_set/SP1/ENCODE+Transcription+Factor+Targets) were retrieved from ENCODE database (https://www.encodeproject.org/), accessed through Harmonizome[90] (https://maayanlab.cloud/Harmonizome/). Reactome database (https://reactome.org/) was employed to calculate the significant pathway enrichment. The complete list of human genes annotated on the GRCh38-hg38 reference genome was retrieved from BioMart Ensembl database (http://www.ensembl.org/biomart/martview). All data are available from the authors upon reasonable request. Source data are provided with this paper.

## Code availability
Custom-made R, bash, and ImageJ macro scripts are available upon request from the corresponding author. Quadparser script was downloaded from https://github.com/dariober/, as indicated by Puig Lombardi et al.[86] G4Hunter algorithm was retrieved from https://github.com/AnimaTardeb/G4Hunter, as indicated by Puig Lombardi et al.[86] The MACS2 code for peak calling of G4-ChIP-seq was retrieved from https://github.com/sblab-bioinformatics/dna-secondary-struct-chrom-lands[19].

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

## Acknowledgements

This work was supported by the Italian Foundation for Cancer Research (AIRC; 21850 to S.N.R.); European Research Council (ERC-2013-CoG; 615879 to S.N.R.); Deutsche Forschungsgemeinschaft (DFG, German Research Foundation; Project-ID 329628492–SFB 1321 and Project-ID 213249687–SFB 1064 to G.S.). Funding for open access charge: Italian Foundation for Cancer Research (AIRC).

## Author contributions

S.L. designed and performed experiments, analyzed data, and wrote the paper; M.N. performed experiments and analyzed data; F.M.C., M.K., and H.D.M. gave technical support and conceptual advice; G.S. and S.N.R. supervised the project and acquired funds; and S.N.R. wrote the paper.

## Competing interests

The authors declare no competing interests.
