## [Peer Review File · Nature Communications]

REVIEWER COMMENTS

Reviewer #1 (Remarks to the Author):

In this manuscript Lago et al. present the prevalence of G4s in liposarcoma cells by means of BG4-ChIP and discuss their potential role in transcription regulation and cell differentiation associating the BG4-ChIP dataset with transcription profiles (RNA-Seq) and chromatin accessibility (ATAC-Seq). The authors find a significant association between G4s detected by BG4-ChIP and actively transcribed genes in accessible regions of chromatin. The authors further corroborate the association between G4s and active transcription by performing Co-I with TFs, confirming co-Immuno precipitation of G4-peaks with TF binding sites, which is an important and novel observation. Although very similar observations were reported in 2016 in Nat Genet for human keratinocytes, it is important to report that G4-association with highly transcribed genes could be recapitulated in different cell lines and that different G4s are detected across different cell lines. In light of the above, I am supportive for publication of this manuscript in Nat Commun.

However, there are some major concerns that needs addressing prior publication:

1) The first concern that I have is on the poor enrichment of oG4 in BG4-ChIP peaks detected in gene promoters. The authors justify this with the high non-canonical nature of most of G4s detected by G4-seq, which is surely a characteristic of the G4s detected in G4-seq. But if this was the case, then they should see similar trends also in non-promoter regions (all genes) and not exclusively in promoters. If there are G4s predicted by Quadparser (so in theory canonical) but not detected by G4-seq, it is likely that these G4s are indeed not folded at all even in vitro so they shouldn't be considered, raising the broader question of why using a predictor when there is an experimentally validated dataset? Furthermore, in the 2016 Nat. Genet. dataset I mentioned above the overlap between BG4-ChIP peaks in promoters and oG4 is >80% so the argument of the non-canonical nature is not sufficient to justify the poor enrichment observed. This should be further discussed and elaborated in the manuscript for clarity of comparison with previous reports. It would also be interesting to analyse how the transcriptional activity of G4-peaks that exclusively overlap with oG4s compare with the rest, maybe those are more highly transcribed?

2) The second concern I have is on the interpretation of the Co-I with TFs data. As I mentioned, this is a novel observation that is important to report, but I have the feeling that the authors are strongly over-interpreting their data. Here the authors state: " These data indicate that TFs AP-1 and SP1 are strictly linked to G4s and suggest that G4s are exploited by cells to facilitate TF interaction with the DNA...", this sentence/conclusion is highly speculative and not supported by the data. Co-I data can be used to state that within the same peaks where a G4 is detected also the TF is detected/localised, which corroborates association of G4s with hotspots of high transcriptional activity. Whether G4-formation facilitates TF binding or is an artefact generated by the high transcriptional activity (e.g. negative supercoiling) cannot be assessed with this data and shouldn't be speculated to avoid sending misleading messages to non expert readers. Similarly, increased transcription due to lack of DNA-methylation should be demonstrated with bisulfite-Seq data and not speculated based on previous reports if the authors wants to claim it in their manuscript

3) The third concern is on the lack of a perturbation control experiment. Most of the conclusion that the authors discussed are based on BG4-ChIP/RNA-Seq/ATAC-Seq on a single cell line and its comparison to previously reported data (Nat Genet 2016). Having a perturbation experiment that confirm the general role of G4s in transcription regulation in liposarcoma cells would be extremely beneficial to supports the authors claims and to generalise the observation reported in the manuscript. For example in the 2016 Nat Genet manuscript, the authors have treated cells with entinostat, a HDAC inhibitor that generated new ATAC-Seq peaks and new G4s, further supporting the positive association of G4s with active transcription in human keratinocytes. It would be valuable to have a similar control in this study, to further strengthen the conclusion and the general concept of G4s as active mark of open chromatin of highly transcribed genes also in liposarcoma cells.

Minor points:

4) The authors focus their analysis on promoters mainly. Most of the BG-ChIP detected are not in promoters and the authors mention that these peaks still are associated with actively transcribed genes but they do not elaborate much on this. It would be good to know more about the distribution of these peaks across the different categories and their relative enrichment with respect to input (rather than percentage across of all peaks that might be misleading). For example, 5UTR peaks seems also very enriched with actively transcribe genes, I was surprised to see that the authors have not elaborated/commented much on this.

5) The definition of G4s is highly confusing across the manuscript (oG4s vs pG4s). Also the authors mention different algorithms (quadparser G4-hunter) so it is not clear which one they are using in the different part of the manuscript. Here I would recommend to use only the oG4s as it is an experimentally validated map that does not rely on prediction and for consistency with previous reports.

6) What dataset have the authors used to define their oG4s? Is it the PDS or the K+ map reported in Chambers et al Nat Biotech 2015? Results might vary quite a lot depending on the dataset used and this should be clarified in the text. For example the >700,000 G4 figures refers to the PDS map

7) In figure 1C and 1E the definition "all genes" makes little sense. This should be non-promoter regions or something similar

8) When discussing the difference of total number of G4s detected between the keratinocytes and liposarcoma cells they speculate that such differences can be related to the different doubling times of the cell lines. Again this is highly speculative and not supported by data and should be removed unless it can be proven with more evidence. Differences measured in G4-prevalence are based empirical observation that are in agreement with the basic hypothesis of G4s playing a role in defying cell identity via transcriptional regulation, therefore there should be no need to invoke potential correlation with doubling times that cannot be proven with the data presented in the manuscript

9) The final sentence about G4 induction by the antibody is redundant and a bit out of context. The authors have strong evidence to support that there is a different distribution of G4s across different cell lines that cannot be obviously justified by induction of these structures, which should be comparable across any cell line. I would remove this sentence that risk to confuse the reader without adding much to an already interesting manuscript.

Reviewer #2 (Remarks to the Author):

The article from Lago and colleges with the title: "promoter G-quadruplexes and transcription factors cooperate to shape the cell type-specific transcriptome" is a short mini-article. They determined the G4 landscape in two different cancer cell lines: HaCaT cells as well as 93T449 cell. They revealed that these cells differ in the regions where G-quadruplexes form. In subsequent experiments they revealed that G4 formation mainly forms within promoters and directly correlates with increased expression rates within these cells.

They speculate that G4 formation supports the transcriptional changes of cancer subtypes. The function of AP-1 and SP1, both transcription factors, are linked to G4 formation within these promoters.

The hypothesis of the manuscript is very interesting but needs further experiments to strengthen their findings and raise to a conclusion. In general I would like to note, that an introduction and discussion is currently not included in the manuscript.

Major concerns:

1. The authors state the G4 formation is different in these tumors and that this is linked to a unique cell specific expression pattern. Can this be used to further classify the tumor? What are the main genes that are affected that drive tumorigenesis in these diseases? Are those also

affected due to G4 formation? meaning could it be that G4 formation supports/stimulates expression patterns that is unique for the tumor?

2. Do the differently expressed genes, controlled by G4 formation, belong to a specific pathway/ e.g. growth factors, angiogenesis etc.? are the pathways, which are altered between the tumors, identical?

3. It is known that G4 cause mutations and deletions in different cells. Recently it was published that G4 formation within breast cancer patients causes mutations and deletions of which can be used to define tumor subtypes. Are the differences in G4 in these tumors also linked to the mutagenic burden of the tumor? Are differently expressed genes linked to the entity? Why have the authors selected these two tumor models? Because these cells do not have any source in common and also have a complete different epigenetic background, it is not surprising that the G4 landscape is different.

4. BG4 antibody has caused problems in the past of unspecific binding. How have the authors validated the antibody?

5. They assume that G4 formation might be different in these cells due to differences in doubling time (Figure S6). Have they considered that it could also be due differences in cell cycle?

6. It is very interesting that there are these changes between different cell types and that there are changes in regulation of transcription. If this is one new "factors" that support cell identity, it would be very rewarding to determine the proteins, epigenetic modification that drives G4 formation. Since G4 formation is different between the cells, either the epigenetic changes within these cells drives (indirectly) G4 formation or a specifically expressed factors for this cell type is specifically targeting those G4s.

7. From their data SP-1 and AP1 are promising candidates that support or bind to G4. Here to get a better picture in vitro data are required that test the effect of both TF on G4 formation. do SP1 and AP1 have different functions or expression levels in these two entities?

8. In general, a summarizing discussion that reflects their data in the context of the research would be highly beneficial. They could discuss previous findings that G4 also down-regulate transcription of specific oncogenes.

Reviewer #3 (Remarks to the Author):

DNA G-quadruplexes (G4) have been related to transcriptional regulation. In this manuscript authors describe their studies using a G4 antibody to identify G4 regions in the genome of a liposarcoma cell line and compare them with data available for keratinocytes. They found that G4 in promoters are associated with high transcription level and that AP1 and Sp1 transcription factor binding sites are highly represented in G4 regions. Their main conclusion is that "G4s and their associated TFs cooperate to determine cell-specific transcriptional programs".

The bioinformatics analysis clearly shows a correlation between G4s and transcript levels. However, I consider that authors are far from fully demonstrating their main claim. The experimental design may not be the most appropriate to address the relevance of G4s in determining cell-specific transcriptional programs and whether G4s are crucial for that. A direct approach would have included the use of one (or ideally more) cell lines at different times during differentiation. Comparing two quite different cell types certainly provides information about their similarities and differences but it is difficult to establish if they arise solely from their G4 status. Also, manipulation of regions that for G4s and analysis of the effect on transcriptional control would be necessary to fully demonstrate the authors' major claims.

Specific comments

1. Line 36. Justify the use of this cell line

2. Do you get similar results using algorithms others than HOMER? Justify the use of this one.

3. Line 38. Validate the anti-G4 antibody and/or provide convenient references.

4. Line 42. Use of "significant". Please confirm that this is accompanied by appropriate stats throughout the manuscript.

5. Line 43. High GC frequency "towards" the ChIP peak center but not "at" the center for G and C.

6. Line 128 (SF6?). Correlating the presence of G4s with a parameter like cell growth, which

integrates many processes, might not be relevant, unless experimentally demonstrated independently.

7. Lines 161-162. This claim goes way too far from being supported by the results. This may require deleting G4s and determine effect on transcription at some individual loci.

8. Correlation with TF binding sites may be related to the high GC content of those sites, not necessarily to the function of these TFs. Please comment.

REVIEWER COMMENTS

Reviewer #1 (Remarks to the Author):

In this manuscript Lago et al. present the prevalence of G4s in liposarcoma cells by means of BG4-ChIP and discuss their potential role in transcription regulation and cell differentiation associating the BG4-ChIP dataset with transcription profiles (RNA-Seq) and chromatin accessibility (ATAC-Seq). The authors find a significant association between G4s detected by BG4-ChIP and actively transcribed genes in accessible regions of chromatin. The authors further corroborate the association between G4s and active transcription by performing Co-I with TFs, confirming co-immunoprecipitation of G4-peaks with TF binding sites, which is an important and novel observation. Although very similar observations were reported in 2016 in Nat Genet for human keratinocytes, it is important to report that G4-association with highly transcribed genes could be recapitulated in different cell lines and that different G4s are detected across different cell lines. In light of the above, I am supportive for publication of this manuscript in Nat Commun.

However, there are some major concerns that needs addressing prior publication:

1) The first concern that I have is on the poor enrichment of oG4 in BG4-ChIP peaks detected in gene promoters. The authors justify this with the high non-canonical nature of most of G4s detected by G4-seq, which is surely a characteristic of the G4s detected in G4-seq. But if this was the case, then they should see similar trends also in non-promoter regions (all genes) and not exclusively in promoters.

To address this comment, we calculated the intersection between pG4s, oG4s and BG4 ChIP-G4s in the previously published dataset of HaCaT cells, obtaining quite similar results to those obtained in LPS cells in the present work (**Supplementary Figure 4A-D**). We inferred that the bias in pG4s and oG4s overlapping can be partially explained by the fact that the majority of ChIP-G4s corresponded to canonical G4 motifs (**Supplementary Figure 2A**), while more than 60% of oG4s contained long loops or non-canonical patterns. It may also indicate that promoter G4s have peculiar features that discriminate them from G4s occurring in other genomic regions or that prevent them to be efficiently detected by G4-seq. In general, we believe that ChIP-G4s are more reliable than oG4s since they are detected in more physiological condition within cells, in their chromatin context and without applying PCR cycling conditions, as it is done instead during G4-seq protocol published by Chambers et al.

These aspects have been added in the main text in the Results section at pag.6 and figures added in Supplementary Material.

If there are G4s predicted by Quadparser (so in theory canonical) but not detected by G4-seq, it is likely that these G4s are indeed not folded at all even *in vitro* so they shouldn't be considered, raising the broader question of why using a predictor when there is an experimentally validated dataset?

It is true that G4-seq is an experimentally validated dataset reporting the human genomic regions that have G4 folding potential *in vitro*. However, G4-seq protocol described by Chambers et al. is still an artificial way to detect G4s that can introduce some biases and potentially also exclude G4s that are instead folded in a different context. In fact, G4-seq relies on the fragmentation of the genome in a random way and detection of DNA polymerases stops, attributed to G4 folding. The folding of G4s in this context is induced by K^+ . These experimental settings may create artifacts in multiple ways: i) fragmentation, even if in principle randomly occurring, can favor detection of some G4s with respect to others; ii) the presence of strong G4s induced by K^+ can stop polymerase and hinder the detection of G4s in adjacent regions; iii) addition of K^+ ions can stabilize G4s in such a way to favor some equilibria that are different from those in the physiological state and thus indirectly hinder detection of less favored G4s. iv) Moreover, the polymerase enzyme used in the Illumina sequencing system has a very high processivity which can lead to resolution of the least stable G4s, missing their detection. v) Similarly, the non-physiological temperatures used during sequencing can affect detection of the least stable G4s. In short, it is not guaranteed that G4s that are folded in a "forced" context and naked DNA, do represent the real physiological G4 landscape. Computationally predicted G4s (pG4s) following a canonical G4 pattern are as well surely not representative of the real G4 landscape. But in light of the above observations, we believe that comparison of ChIP-G4s with both datasets is a correct and more robust analysis than only comparison with oG4s. Finally, several recent papers still refer to the computationally predicted G4 landscape and thus we consider it worth to also keep those datasets as comparison.

Furthermore, in the 2016 Nat. Genet. dataset I mentioned above the overlap between BG4-ChIP peaks in promoters and oG4 is >80% so the argument of the non-canonical nature is not sufficient to justify the poor enrichment observed. This should be further discussed and elaborated in the manuscript for clarity of comparison with previous reports. It would also be interesting to analyse how the transcriptional activity of G4-peaks that exclusively overlap with oG4s compare with the rest, maybe those are more highly transcribed?

In the 2016 Nat. Genet. paper, overlaps between oG4s and ALL BG4 ChIP-G4s are shown. We have now calculated and reported the overlap between the different sets of G4s also for the published HaCaT cells dataset.

Consistently with what observed in 93T449 cells, there is a strong (more than 95%) overlap between oG4s, pG4s and ChIP-G4s when considering all G4-containing genes (**Supplementary Figure 3A left panel**); in contrast, when considering promoters only, about 23% and 100% of ChIP-G4s correspond to the oG4s and pG4s dataset, respectively (**Supplementary Figure 3A central panel**). Considering the group of non-promoter BG4 ChIP G4s only, we found an overlap with oG4s and pG4s higher than 97% (**Supplementary Figure 3A, right panel**). These results are a strong indication that promoter G4s have peculiar features that discriminate them from G4s in other genomic regions.

The transcriptional output of ChIP-G4 genes has been evaluated for 93T449 cells (**Supplementary Figure 5B**) comparing the expression of genes with oG4s, ChIP-G4s, the overlap of genes with both ChIP-G4s and oG4s (ChIP-G4 and oG4) or genes that are outside of the overlapping (ChIP-G4 not oG4). Furthermore, these genes were grouped in order to consider them independently of the G4 position (ALL), the ones with G4s at their promoter (PROMOTER) or elsewhere than the promoter (NON PROMOTER). As already described in the manuscript, we noticed that globally ChIP-G4 genes have higher transcript abundance with respect to oG4s, especially when the G4 is in the promoter. A similar trend is observed for oG4s, even if this is visually less evident. When comparing genes that have G4s both according to ChIP-seq and G4-seq (i.e. ChIP-G4s and oG4s), the difference of expression between G4 categories (ALL, PROMOTERS and non-PROMOTERS) is much less evident, suggesting that G4s identified by the G4-seq strategy may be restricted to a subset of G4s with peculiar features that make them easier to be captured by this strategy, independently from gene transcription. Hypothetically those are the most thermodynamically stable G4s, as previously discussed. Finally, when looking at G4s that are identified by ChIP-seq but not detected by G4-seq (ChIP-G4, not oG4s), the trend observed for ChIP-G4s and oG4s is maintained, but not statistically significant, probably due to the low numerosity of the groups. We can therefore infer that G4s identified by G4-seq are not completely representative of the cell specific G4-landscape, as instead ChIP-G4s are. Moreover, it can be hypothesized that G4s identified by G4-seq have slightly different structures from the ones that are folded within cells in a more physiological context and observed by ChIP-seq.

These aspects have been added at pages 6 and 7 and figures added in the Supplementary material.

2) The second concern I have is on the interpretation of the Co-I with TFs data. As I mentioned, this is a novel observation that is important to report, but I have the feeling that the authors are strongly over-interpreting their data. Here the authors state: " These data indicate that TFs AP-1 and SP1 are strictly linked to G4s and suggest that G4s are exploited by cells to facilitate TF interaction with the DNA...", this sentence/conclusion is highly speculative and not supported by the data. Co-I data can be used to state that within the same peaks where a G4 is detected also the TF is detected/localised, which corroborates association of G4s with hotspots of high transcriptional activity. Whether G4-formation facilitates TF binding or is an artefact generated by the high transcriptional activity (e.g. negative supercoiling) cannot be assessed with this data and shouldn't be speculated to avoid sending misleading messages to non expert readers.

As the reviewer suggests, the hypothesis of cooperativeness or independency of G4s and TFs has been more explicitly discussed in the main text, highlighting the fact that the presented data indicate the ability of TFs to bind to G4 folded regions, but that they cannot indicate whether G4s actively contribute to TFs binding or their presence is just a consequence of high transcription rates. Both hypotheses have been stated in the manuscript discussion: *"These data corroborate the observation that G4s and TFs AP-1 and SP1 are strictly linked to transcriptionally active regions. The co-occurrence of G4s and TFs could be: i) independent and due to G4 stimulating conditions typical of highly transcriptionally active DNA regions (such as negative supercoiling) or ii) they can have a cooperative role in which G4s are exploited by cells to facilitate TF interaction with the DNA: this effect can be reached by G4-mediated exposure of the binding region or altered DNA methylation state, as G4s were reported to prevent CpG island methylation, condition which is required for transcription initiation"*.

Similarly, increased transcription due to lack of DNA-methylation should be demonstrated with bisulfite-Seq data and not speculated based on previous reports if the authors wants to claim it in their manuscript

We have now discussed the potentially cooperative role between G4s and stimulation of transcription in a more hypothetical way and used previous reports showing the lack DNA-methylation at G4 sites as support of the hypothesis. It is now clearer that this observation is only used as theoretical background, based on previous experimental data, that can guide future analysis for a better understanding of how G4s are involved in the regulation of highly transcribed genes.

3) The third concern is on the lack of a perturbation control experiment. Most of the conclusion that the authors discussed are based on BG4-ChIP/RNA-Seq/ATAC-Seq on a single cell line and its comparison to previously reported data (Nat Genet 2016). Having a perturbation experiment that confirm the general role of G4s in transcription regulation in liposarcoma cells would be extremely beneficial to supports the authors claims and to generalise the observation reported in the manuscript. For example in the 2016 Nat Genet manuscript, the authors have treated cells with entinostat, a HDAC inhibitor that generated new ATAC-Seq peaks and new G4s, further supporting the positive association of G4s with active transcription in human keratinocytes. It would be

valuable to have a similar control in this study, to further strengthen the conclusion and the general concept of G4s as active mark of open chromatin of highly transcribed genes also in liposarcoma cells.

As suggested by the reviewer, we have now performed perturbation analysis to confirm the positive association of G4s with active transcription in liposarcoma cells. Specifically, we detected G4s foci in cells treated either with the HDAC inhibitor entinostat, stabilizing transcriptionally active chromatin, or the RNA-pol II inhibitor actinomycin D, preventing transcriptional elongation of the RNA chain [[doi:10.1073/pnas.82.16.5328](https://doi.org/10.1073/pnas.82.16.5328)] (**Figure 3**). We observed high and dose-dependent increase in G4 foci in liposarcoma cells upon entinostat treatment (**Figure 3A and B, upper panel**), confirming previous results obtained on keratinocytes (Nat Genet 2016) and indicating that folded G4 structures, and not the corresponding G-rich sequences, associated with transcriptionally permissive chromatin regions. On the other hand, folded G4s foci in liposarcoma cells decreased upon actinomycin D treatment (**Figure 3A and B, lower panel**) supporting the connection between transcription and G4s formation. These observations have been added in the main text at pp. 8-9.

Minor points:

4) The authors focus their analysis on promoters mainly. Most of the BG-ChIP detected are not in promoters and the authors mention that these peaks still are associated with actively transcribed genes but they do not elaborate much on this. **It would be good to know more about the distribution of these peaks across the different categories and their relative enrichment with respect to input (rather than percentage across of all peaks that might be misleading).** For example, **5'UTR peaks seems also very enriched with actively transcribe genes, I was surprised to see that the authors have not elaborated/commented much on this.**

As suggested by the reviewer, we have now added a figure showing the relative enrichment of genomic functional regions containing G4 peaks (i.e. Promoter, 5'UTR, 3'UTR, TTS etc.) with respect to input, where the input is the complete human genome (**Supplementary Figure 4A**). Relative enrichment was calculated as (functional region frequency in G4-ChIP peaks - functional region frequency in whole genome)/ functional region frequency in whole genome. Enrichment of 0 means that the frequency of the functional region in the human genome and in ChIP-peaks is the same, which is equivalent to a random representation. Negative enrichment corresponds to under-representation, while positive enrichment to over-representation. From these results, promoter regions stand out as the most prominently enriched category of functional elements containing G4 peaks. A positive enrichment is also visible for 5'UTRs, despite to a much lower extent with respect to promoters. All the other evaluated functional elements showed instead a slight underrepresentation of G4 structures.

5'UTR are regions that partially overlap with gene promoters. In particular, they commonly span the genomic region going from the TSS to the ATG of first exon. This first observation can generate a bias in the association of G4s with this functional region since the resolution of G4 peaks obtained by ChIP-seq is of about 150-200 bp. Therefore, G4s mapping to the 5'UTR should be treated carefully.

Another observation regards the well-documented presence of Transcription Factor Binding Sites (TFBSs), as well as other transcriptional regulatory elements, splicing sites and structural regulatory modulators in 5'UTR [[doi: 10.1007/s00018-012-0990-9](https://doi.org/10.1007/s00018-012-0990-9); [doi: 10.1093/nar/gkx751](https://doi.org/10.1093/nar/gkx751)].

Combining these two observations it is not surprising that genes with 5'UTR G4s are transcribed to a high level. This fact has been now commented in the main text at page 3.

5) The definition of G4s is highly confusing across the manuscript (oG4s vs pG4s). Also the authors mention different algorithms (quadparser G4-hunter) so it is not clear which one they are using in the different part of the manuscript. Here I would recommend to use only the oG4s as it is an experimentally validated map that does not rely on prediction and for consistency with previous reports.

pG4s and oG4s are defined in the main text at pag. 5: "We next compared the amount of ChIP-G4s to that of putative G4s (pG4s) calculated by Quadparser and of "observed G4s" (oG4s), i.e. genomic regions previously observed to stop polymerase progression in vitro 23". We clarified in the main text what the definition of pG4s indicated in the different parts: "The presence of G4s within ChIP peaks was assessed through Quadparser and G4Hunter and the predicted putative G4s referred to as pG4s." (pag.2), and "We next compared the amount of ChIP-G4s to that of putative G4s (pG4s) computationally calculated by Quadparser (loop 0-12) and of "observed G4s" (oG4s), i.e. genomic regions previously observed to stop polymerase progression in vitro in the presence of K+23. From now on, pG4s refer to Quadparser (loop 0-12) predicted G4s, it was preferred among the employed tools for G4 prediction since it was able to predict the highest number of pG4s." (pag.5).

As previously discussed (point 1), we decided to keep as a reference both the computationally predicted (pG4s) and experimentally observed (oG4s) for two main reasons: i) Much of the recent literature still uses computational prediction tools to predict G4 folding; ii) the G4-seq strategy, by which the oG4 dataset is generated, is still an artificial way to detect G4s, that works on naked DNA and non-physiological conditions. Therefore, we deem that considering both datasets provides a more complete view of the G4 detection potential by different techniques.

6) What dataset have the authors used to define their oG4s? Is it the PDS or the K+ map reported in Chambers et al Nat Biotech 2015? Results might vary quite a lot depending on the dataset used and this should be clarified in the text. For example the >700,000 G4 figures refers to the PDS map

The used oG4s datasets is the one in the presence of K⁺, since it is a condition closer to the physiological one when compared to G4-ligand (PDS)-mediated induction of G4 folding.

As correctly noted by the reviewer, we now specified in the main text the used dataset: "We next compared the amount of ChIP-G4s to that of putative G4s (pG4s) computationally calculated by Quadparser (loop 0-12) and of "observed G4s" (oG4s), i.e. genomic regions previously observed to stop polymerase progression in vitro in the presence of K⁺" pag 5; and added a reference in the *Data availability* section: "The oG4s dataset produced in the presence of K⁺ was retrieved from GEO at the following accession number GSE63874."

7) In figure 1C and 1E the definition "all genes" makes little sense. This should be non-promoter regions or something similar.

The definition "all genes" in figure 1C and E actually refers to all genes (both genes with promoter G4s and non-promoter G4s). It was specified in Fig 1C, to make clear that we did not exclude non-expressed genes from the analysis, but we considered *all genes*, and in Fig 1E to show the differences of G4 number when *all genes* with identified G4s or *genes with promoter G4s* only were considered. Data represented in figure 1E have now been investigated deeper, and the panel has been moved to **Supplementary Figure 3B and C**. The *non-promoter* category, containing all genes that contain G4s elsewhere than promoter regions has been added for comparison both for 93T449 and HaCaT cells.

8) When discussing the difference of total number of G4s detected between the keratinocytes and liposarcoma cells they speculate that such differences can be related to the different doubling times of the cell lines. Again this is highly speculative and not supported by data and should be removed unless it can be proven with more evidence. Differences measured in G4-prevalence are based empirical observation that are in agreement with the basic hypothesis of G4s playing a role in defying cell identity via transcriptional regulation, therefore there should be no need to invoke potential correlation with doubling times that cannot be proven with the data presented in the manuscript

Data on doubling times have been removed as suggested.

9) The final sentence about G4 induction by the antibody is redundant and a bit out of context. The authors have strong evidence to support that there is a different distribution of G4s across different cell lines that cannot be obviously justified by induction of these structures, which should be comparable across any cell line. I would remove this sentence that risk to confuse the reader without adding much to an already interesting manuscript.

The final sentence about G4 induction by the antibody has now been removed from the main text.

Reviewer #2 (Remarks to the Author):

The article from Lago and colleagues with the title: “promoter G-quadruplexes and transcription factors cooperate to shape the cell type-specific transcriptome” is a short mini-article. They determined the G4 landscape in two different cancer cell lines: HaCaT cells as well as 93T449 cell. They revealed that these cells differ in the regions where G-quadruplexes form. In subsequent experiments they revealed that G4 formation mainly forms within promoters and directly correlates with increased expression rates within these cells. They speculate that G4 formation supports the transcriptional changes of cancer subtypes. The function of AP-1 and SP1, both transcription factors, are linked to G4 formation within these promoters.

The hypothesis of the manuscript is very interesting but needs further experiments to strengthen their findings and raise to a conclusion. In general I would like to note, that an introduction and discussion is currently not included in the manuscript.

Introduction, Discussion and Conclusion sections have been added to the manuscript.

Major concerns:

1. The authors state the G4 formation is different in these tumors and that this is linked to a unique cell specific expression pattern. Can this be used to further classify the tumor? What are the main genes that are affected that drive tumorigenesis in these diseases? Are those also affected due to G4 formation? meaning could it be that G4 formation supports/stimulates expression patterns that is unique for the tumor?

To understand if the G4 landscape characteristic of the two cell lines can be used to classify the tumor, we performed a pathway analysis on the G4-containing genes that are unique for the two cell lines (i.e. selecting only those genes which displayed folded G4s only in 93T449 or in HaCaT cells line). The R package “ChIPpeakAnno” [doi: 10.1186/1471-2105-11-237; doi: 10.1007/978-1-62703-607-8_8] combined with Reactome database (<https://reactome.org/>) was used to calculate the enrichment of specific pathways on the only basis of cell line specific G4s (**Supplementary Figure 8A**). The most represented pathways identified in the two cell lines are different: in particular, immune system-related pathways are prevalent in 93T449 LPS cells, while vesicle-mediated, membrane trafficking, hemostasis and cell growth signals prevail in HaCaT cells. Despite the different pathways can be reconducted to the specific cell line [doi: 10.7759/cureus.3549, doi: 10.1097/PAS.0b013e31824f2594, doi: 10.1097/0000478-199708000-00002, doi: 10.1111/jdv.15859] they are not unique for a specific tumor or cell type and they alone are not sufficient for tumoral/cellular classification. In addition, comparison of the two different cell lines is not sufficient for the identification of tumorigenesis associated pathways, as the corresponding normal cells, like normal adipocytes for 93T449 and normal keratinocytes for HaCaT, would be necessary for a meaningful comparison.

Considering the two cell lines separately, HaCaT cell line is a spontaneously immortalized, non-tumorigenic cell line, the immortalization of which is caused by a p53 mutation that inhibits p53 capacity to regulate DNA repair mechanisms, and in parallel the loss of chromosome 3p arm, which carries fundamental genes for senescence. Combination of such alterations, together with increased telomerase activity are considered the main causes of HaCaT cells immortalization, which is the first step towards tumorigenesis [doi: 10.1002/(sici)1098-2744(199811)]. To understand if telomerase hyperactivation could be linked to G4 formation, we searched for G4-peaks in telomerase-associated genes in HaCaT cells. Interestingly, we found 20 G4 peaks in important genes encoding for telomerase itself or other components of its regulatory machinery (i.e., TERC, TERT, TERF1, TERF2, TEP1, ACD, RTEL1 and POT1). Importantly, none of these genes was found to fold into G4 structures in the normal keratinocyte control cell line nHEK [doi: 10.1038/ng.3662]. The latter result highlights the involvement of G4 structures in supporting/stimulating tumorigenic expression patterns linked to the cellular entity.

It is more complex to evaluate G4 role in 93T449 cells, due to the lack of information on normal adipocytes and on the limited knowledge of the exact mechanisms leading to tumorigenesis. The main altered genes in 93T449 cells are MDM2 and CDK4, which are overexpressed in WDLPS. Both genes display folded G4s at their promoter according to BG4-ChIP seq data (**Figure 2A, Supplementary File 1**) (doi: 10.1093/nar/gkaa1273). Both genes are genetically amplified in the cell line [doi: 10.1016/j.canlet.2008.08.025], constituting a supernumerary giant ring chromosome, therefore we cannot reconduct their overexpression and involvement in tumorigenesis uniquely to the presence of G4s, but according to our data, G4s likely contribute to keep the sustained expression of the genes.

These aspects have been added to the main text at pp 13-14 and in the discussion.

2. Do the differently expressed genes, controlled by G4 formation, belong to a specific pathway/ e.g. growth factors, angiogenesis etc.? are the pathways, which are altered between the tumors, identical?

The altered pathways between the two cell lines are different since HaCaT cells immortalization is driven by p53 mutation, loss of senescence genes encoded on chromosome 3p and hyperregulated telomerase activity. 93T449 cells tumorigenesis is instead mainly driven by MDM2 and CDK4 overexpression, resulting in suppression of p53 activity and stimulation of cell cycle progression. Thus, different pathways lead in the end to similar tumorigenic outcomes sustained by cell death evasion, illimited duplication potential and genomic

instability due to impaired DNA repair mechanisms. For this reason, we expect that comparison of differentially expressed genes between the two cell lines highlights cell-type specific pathways in addition to differences in the tumorigenic induction. We calculated the enriched pathways considering differentially expressed genes, controlled by G4 in the two cell lines by mean of the R package “ChIPpeakAnno” [doi: 10.1186/1471-2105-11-237; doi: 10.1007/978-1-62703-607-8_8] combined with Reactome database (<https://reactome.org/>) (**Supplementary Figure 8B**). In line with the known features of HaCaT cells, we found enrichment in genes involved in fibroblast growth factor receptors (FGFR), typical of self-sufficiency in growth factor, cytochromes P450 related genes, which are a group of enzymes involved in skin protection, with some of them activated as defense against UVA-radiation oxidative damage [doi: 10.1046/j.1365-2133.2001.04490.x, doi: 10.1016/j.tiv.2009.12.023, doi: 10.1124/dmd.111.042085], DNA strand elongation, VEGFR vascular permeability and telomere synthesis, all pathways that are typically considered cancer hallmarks [doi: 10.1016/j.cell.2011.02.013].

93T449 displayed instead enriched G4-driven pathways involved in Immune system, suggesting a key role in the mechanisms of immune system evasion defense. Specifically, we found over representation of genes of the Toll-like receptors 7/8 response that are involved in the production of immunosuppressive cytokines, increased cell proliferation and resistance to apoptosis [<https://doi.org/10.1038/sj.onc.1210913>], and Toll-like receptor 9, which is typically stimulated when the immune system responds to cancer-dependent inflammation [doi: 10.2147/OTT.S174274, doi: 10.3389/fimmu.2014.00330]. Besides, we found enrichment of O-linked glycosylation pathway, the activation of which is necessary for adipocytes differentiation [doi: 10.1016/j.bbrc.2010.06.105] and organelle biogenesis and maintenance: high organelle dynamic renewal, especially that of mitochondria, is indeed typical of adipose cells [doi: 10.1080/21623945.2019.1574194].

In summary, these results, support that G4s and gene expression are strictly connected and associated to the stimulation/establishment of cell type and tumor associated molecular pathways.

These aspects have been discussed in the main text at pp 12-14 and in the discussion.

3. It is known that G4 cause mutations and deletions in different cells. Recently it was published that G4 formation within breast cancer patients causes mutations and deletions of which can be used to define tumor subtypes. Are the differences in G4 in these tumors also linked to the mutagenic burden of the tumor? Are differently expressed genes linked to the entity?

We added an association study to understand if 93T449 cell-specific G4s are associated to pan-cancer mutations.

*“A recent work by R.H.H. et al. [doi: 10.1038/s41588-020-0672-8.] supported by previous literature [Cell145, 678–691 (2011), Nat. Genet.31, 405–409 (2002), Genome Res.28, 1264–1271 (2018), Nat. Biotechnol.33, 1–7 (2015).] showed that pan-cancer somatic variants (SVs), cancer-related Copy Number variants (CNVs), i.e. amplifications, and Single Nucleotide Variants (SNVs) are enriched in breast cancer-specific G4s. To test if WDLPS-specific G4s were associated to cancer variants, we calculated the overlap between 93T449 cell-specific CNVs or SVs [10.1534/genetics.117.300552] and G4 ChIP peaks found in 93T449 or HaCaT cells. Interestingly, we found that 73% of 93T449 CNVs (299 out of 411 detected CNVs) overlapped with folded G4s peaks, and 47% of all CNVs (193 out of 411) were in genomic regions that harbor 93T449 cell line-specific G4s (i.e., G4s that are not folded in the HaCaT cell line). On the contrary, only 1% of all CNVs colocalized with HaCaT-specific G4s (**Supplementary Figure 8C and D**). This result supports a meaningful association between G4s and pan-cancer CNVs, as also previously observed [Nat. Biotechnol.33, 1–7 (2015)]. Different results were found instead for SVs (Insertions, deletions, inversions, intra- and inter-chromosomal translocations), the abundance of which in cell line-specific G4s was below 3%, thus indicating no correlation (**Supplementary Figure 8E**). Comparing 93T449 genes affected by CNVs and harboring cell line specific-G4s with the same genes in HaCaT cells, we did not see any significant difference in the general expression level. The latter result suggests that, while G4s are associated with pan-cancer genomic instability regions, their role in the control of gene expression works through a different mechanism, which could be recruitment of TFs.”*

Data availability and analysis procedure are described in the materials and methods section, and results have been presented in the Result section at pages 13-14, in the Discussion and **Supplementary Figure 8C-E** have been added in Supporting Material.

Why have the authors selected these two tumor models? Because these cells do not have any source in common and also have a complete different epigenetic background, it is not surprising that the G4 landscape is different.

As the reviewer highlighted, in the present manuscript we compared two very different tumor models: HaCaT are non-malignant immortalized keratinocytes, while 93T449 are Well-differentiated Liposarcoma cells. Importantly, the two cell lines belong to the two main types of differentiated cells in vertebrates: epithelial and mesenchymal respectively. The choice of such distant cell types, with complete different epigenetic background, is highly suitable for the purpose to define G4s involvement in the establishment/maintenance of cell-specific transcriptome. So far, genome wide comparisons of in-cell detected G4s was very limited (doi: 10.1002/bies.201900091, doi: 10.1038/s41594-018-0131-8), therefore there is no obvious reason to link the epigenetic background to a different G4 landscape. With this premise, the employment of epigenetically different

cell lines was the best choice to robustly highlight a parallel variation of folded G4s in dependence of the cell transcriptome and epigenome. Had we selected more similar cell types, we would have likely observed less significant G4 landscape variations, which would have been of more difficult interpretation. Thus, in the absence of previous clear proof that G4 folding is modified in parallel with the epigenome, the choice of cell types with intrinsic epigenetic differences looked to us the most correct way to assess G4 role.

4. BG4 antibody has caused problems in the past of unspecific binding. How have the authors validated the antibody?

Each used batch of purified BG4 antibody was validated *in vitro* by mean of native electrophoretic mobility shift assay (EMSA). A dose dependent binding intensity was measure for BG4 when incubated in the presence of three human genome G4-folded sequences (c-myc, bcl.2 and c.kit), while no binding was detected in the presence of a G-rich non-G4 single stranded sequence (scrambled). One representative EMSA experiment has been added in **Supplementary Figure 1A** and described in the material and methods section as well as in the main text: "BG4 antibody was purified and its ability to discriminate G4 structures over non folded single stranded sequences was validated *in vitro* by EMSA assay." References of papers employing the same strategy and obtaining comparable results in BG4 activity validation were mentioned in the main text: doi: 10.1038/NCHEMBIO.2228, doi: 10.1371/journal.pone.0158794

5. They assume that G4 formation might be different in these cells due to differences in doubling time (Figure S6). Have they considered that it could also be due differences in cell cycle?

We have removed the doubling times analysis and inference as suggested by other referees.

6. It is very interesting that there are these changes between different cell types and that there are changes in regulation of transcription. If this is one new "factors" that support cell identity, it would be very rewarding to determine the **proteins, epigenetic modification that drives G4 formations**. Since G4 formation is different between the cells, **either the epigenetic changes within these cells drives (indirectly) G4 formation or a specifically expressed factors for this cell type is specifically targeting those G4s**.

As the reviewer correctly noted, the differences in the G4-landscape between cells are interdependent with epigenetic modifications and the specific expression of regulatory proteins. The present manuscript focuses on how G4s are associated with the cell-specific transcriptome and proposes G4s as players in the establishment of cell identity by mean of the specific recruitment of TFs. Understanding which are the proteins or epigenetic modifications that directly or indirectly drive G4 formation represents another level or research which lies outside the main aim of the present manuscript, but that is surely valuable for future investigation.

Besides, there is an increasing interest and accumulating literature investigating the role of G4s as epigenetic modulators. One of the aspects which has been most widely studied is the relationship between G4s and DNA methylation at CpG islands (CGI). Methylation of CGI on gene promoters is generally recognized as a transcriptional repressive epigenetic modification, since CpG binding proteins recruit the enzymatic machinery necessary to establish silent chromatin [doi: 10.3390/molecules23040944]. Many G4-forming sequences within the human genome harbor CpG sites and are specifically enriched at CpGs with low methylation with respect to CpGs with high methylation [doi: 10.1039/COMB00009D]. Moreover, G4s formed at hypomethylated and open chromatin sites possess higher thermal stability with respect to G4s occurring at highly methylated CGI [doi: 10.1111/febs.15065]. Hypomethylated G4 sites are also enriched in DNA methyltransferase 1 (DNMT1) occupancy and exhibit enhanced affinity for DNMT1 binding, sequestering it and protecting the nearby CGI from methylation [doi: 10.1038/s41594-018-0131-8].

Despite under-represented, G4s may also form in methylated CGI regions (e.g. BCL2 [doi: 10.1016/j.bbrc.2012.12.040], FMR1 and C9orf72 repeats [doi: 10.1073/pnas.91.11.4950, doi: 10.1093/nar/gkv1008], VEGF [doi: 10.1021/acs.analchem.6b00982] and MEST [doi: 10.1371/journal.pone.0113955]). It has been shown that cytosine methylation within the G4 motif can alter G4 thermal stability and topology thereby affecting its ability to interact with VEGF165 and Sp1 proteins. Specifically, DNA methylation at G4 loci causes modification of G4 topology which in turn results in changes in the binding affinity of Sp1, with consequent downstream effect on transcription regulation [doi: 10.1016/j.bbrc.2012.12.040, doi: 10.1371/journal.pone.0113955, doi: 10.3390/molecules23040944]. This is particularly relevant when considering that SP1 is ubiquitously expressed to not only maintain basal transcription of housekeeping genes, but also regulate tissue-specific gene expression [doi: 10.1016/j.bbrc.2008.03.074].

In light of these observations, the expression of DNMT1 and the methylated state of DNA at or close to G4 sites is fundamental to define G4s folding and topology, and in turn, enhance or disfavor binding of TFs such as Sp1.

On the other hand, it was observed a correlation between active chromatin histone marks (such as H3K4me3 and H3K27ac) and G4 formation. Inhibition of HDAC with entinostat compound induced chromatin relaxation, stabilization of transcriptionally active regions and increment of folded G4s structures in both 93T449 (**Figure 3**) and HaCaT cells [doi: 10.1038/ng.3662]. This result suggests that, besides DNA methylation, also histone

modifications contribute to establish the cell-specific G4 landscape. [doi: 10.1038/ng.3662] In addition, G4 induced stabilization by G4-ligands induces heritable epigenetic changes in vertebrate cell lines, by triggering inactivation of the locus in two steps, first the loss of promoter H3K4me3, which subsequently leads to histone H3K9 and DNA cytosine methylation and complete shutdown of expression [doi: 10.1038/NCHEM.2828]

We therefore believe that G4s, epigenetic regulators and TFs influence each other in a complex regulatory mechanism, resulting in the definition of cell identity.

These aspects have been added in the main text at page 15 of Results section and in the Discussion.

7. From their data SP-1 and AP1 are promising candidates that support or bind to G4. Here to get a better picture in vitro data are required that test the effect of both TF on G4 formation.

We have tested the c-myc promoter G4 forming oligonucleotide for its binding to SP1 by CD. This oligonucleotide contains the SP1 binding site, in a single-stranded form. Sp1 is a ubiquitous TF that binds to known double-stranded consensus sequences. It was however demonstrated that Sp1 does also recognize and interact with genomic regions that fold into G4s with the minimal Sp1 consensus motif present or absent: one example of the latter is the G4 of the c-kit oncogene promoter [doi:10.1093/nar/gkr882]. Additional evidence suggested that Sp1-induced transcriptional activity was only possible when the interaction site on the G4 was free from the binding of other G4-binding proteins or ligands, which have instead transcriptional repressory functions and prevent Sp1 binding [doi:10.1038/onc.2014.65, doi: 10.1177/1947601910377493].

In our case, we did not find any obvious interaction in vitro between c-myc G4 and SP1. To note, however, that our data indicate binding in a region close to the G4 (\pm 50.bases from the G4), thus assessing the interaction *in vitro* is not straightforward as the single G4s and their flanking regions would need to be considered each time.

Less investigated is AP-1 TF binding in relation to G4 structure. The connection between G4s and AP-1 is indeed less noticeable given the low G-richness of AP-1 consensus motif. The regulatory activity of AP-1 on several human oncogene promoters with G4 forming potential (like bcl-2 and c-myc) was reported with transcriptional activatory/repressory activity depending on the Jun/Fos dimer composition [doi: 10.1158/1541-7786.MCR-10-0105, Eliopoulos et al. INTERNATIONAL JOURNAL OF ONCOLOGY 2: 883-888, 1993, doi: 10.1177/002215540305101204]

do SP1 and AP1 have different functions or expression levels in these two entities?

The different expression levels of Sp1 and AP-1 TF components between the two cell lines have been evaluated (**Supplementary Figure 10B**) and discussed in the results section:

*“Sp1 and AP-1 are two master TFs which regulate many fundamental processes like cell differentiation, growth, apoptosis, immune and DNA damage response, and chromatin remodelling [doi: 10.3390/ijms21031153, doi: 10.3389/fmicb.2017.02686]. AP-1 TF is a dimeric complex composed of proteins belonging to the Jun (c-Jun, JunB, JunC), Fos (c-Fos, FosB, Fra1 and Fra2), ATF (ATF1-4, ATF-6, b.ATF, ATFx) and Maf family (c-Maf, MafA, MafB, MafG/F/K and Nrl). AP1 activity varies according to the cell type and is modulated through its dimer composition, which is determined by the differential expression of its components and through the sequence of the available AP1 binding sites [doi: 10.3389/fmicb.2017.02686, doi: 10.1242/jcs.01589]. Sp1 activity is instead regulated by post-translational modification and when overexpressed is generally considered a negative prognostic factor for cancer [doi: 10.1016/j.pharmthera.2015.05.008]. To understand if there is an association between G4s and the activity of these two TFs, we looked at the differential expression of their components in 93T449 and HaCaT cells. While Sp1 did not show any significant variation in expression, several of the AP-1 TF components were differentially expressed between the two cell lines (**Supplementary Figure 10B**). This observation supports the interconnection between G4s and TFs, since the different composition of AP-1 dimers reflects its different activity and affinity of for binding sites and possibly also the modulation of different cell-line specific G4s.”*

These aspects have been added in the main text at pages 16-17 of results section.

8. In general, a summarizing discussion that reflects their data in the context of the research would be highly beneficial. They could discuss previous findings that G4 also down-regulate transcription of specific oncogenes.

Introduction and Discussion sections have now been added in the manuscript to explain the data in the context of research and highlight more clearly the importance of the present findings.

Reviewer #3 (Remarks to the Author):

DNA G-quadruplexes (G4) have been related to transcriptional regulation. In this manuscript authors describe their studies using a G4 antibody to identify G4 regions in the genome of a liposarcoma cell line and compare them with data available for keratinocytes. They found that G4 in promoters are associated with high transcription level and that AP1 and Sp1 transcription factor binding sites are highly represented in G4 regions. Their main conclusion is that "G4s and their associated TFs cooperate to determine cell-specific transcriptional programs".

The bioinformatics analysis clearly shows a correlation between G4s and transcript levels. However, I consider that authors are far from fully demonstrating their main claim. The experimental design may not be the most appropriate to address the relevance of G4s in determining cell-specific transcriptional programs and whether G4s are crucial for that. A direct approach would have included the use of one (or ideally more) cell lines at different times during differentiation. Comparing two quite different cell types certainly provides information about their similarities and differences but it is difficult to establish if they arise solely from their G4 status. Also, manipulation of regions that form G4s and analysis of the effect on transcriptional control would be necessary to fully demonstrate the authors' major claims.

As the reviewer highlighted, the comparison of the two different cell lines of Well-differentiated Liposarcoma (93T449) and the immortalized keratinocytes (HaCaT) provides a background of very different epigenetic landscapes. It is exactly this intrinsically different background that we intended to exploit to assess how the G4 landscape is modulated, and compare it to the epigenome and transcriptome, thus contributing to the definition of cell identity. We never stated, nor believed that the cell G4 status is the sole condition to establish cells similarities and differences. Due to the lack of previous analysis on the modulation of the G4s landscape in specific cell types, we started addressing this point from one extreme condition: the comparison of completely different cell lines, the first of mesenchymal origin (93T449) and the second of epithelial origin (HaCaT), which are the two main types of differentiated cells in vertebrates. We believe that this comparison and the obtained results, showing a consistent variation of the genome-wide G4 landscape, are a robust premise for the finer determination of G4 involvement or correlation with the establishment/maintenance of the cell-type specific transcriptome. Without the indications given by the experiments and comparisons performed in the present manuscript, it would be much more complex to assess whether less prominent differences in transcriptome/epigenome acquired during the differentiation of a single cell line are linked to G4s or not.

Specific comments

1. Line 36. Justify the use of this cell line

The reasons for using 93T449 cell line in the present work were now explained in the manuscript. In the introduction section: "*In the present work we applied G4 ChIP-seq to map the folded G4s in cells of well-differentiated liposarcoma (WDLPS), a malignant neoplasia affecting also young adults and children, that is resistant to the current chemotherapies and has a largely unexplored biological background 40–42.*"

In the results:

"The two cell lines belong to the main differentiated cell types in human: mesenchymal and epithelial, respectively. Comparison of cells of two different origins allows to robustly assess genome-wide differences in G4 landscape, accounting for an intrinsically different epigenetic/transcriptional background."

And in the discussion:

"Comparison of two very different cell lines 93T449 and HaCaT was chosen as extreme condition to better highlight cell-type specific differences in the G4 landscape. The consistently different transcriptional and epigenetic states, intrinsic of the two cell lines, were a necessary premise to robustly associate G4 differences with the cell identity, information which was not obvious from the current literature [doi: 10.1002/bies.201900091,doi: 10.1038/s41594-018-0131-8]."

2. Do you get similar results using algorithms others than HOMER? Justify the use of this one.

Peak calling analysis was performed in parallel by mean of Homer software and MACS2 tool, the second was the tool originally used by Hänsel-Hertsch et al. 2016 Nat. Genetics for BG4-ChIPseq peak calling and was therefore applied with the same parameters as described at the URL: <https://github.com/sblab-bioinformatics/dna-secondary-struct-chrom-lands> [Hansel-Hertsch et al 2016]. The obtained results are displayed in **Supplementary Figure S1C**, showing Venn Diagrams of peaks comparison for both 93T449 and HaCaT cells. The peaks identified by either Homer and MACS2 are the 86% and 89% respectively, showing a good similarity in the results obtained by the different tools and high confidence in the reliability of the identified peaks. We chose

Homer identified peaks for all the downstream analysis since it was able to catch a higher number of peaks, that are reliable when visually inspected. Example regions of confident peaks identified by Homer but not by MACS2 are shown in **Supplementary Figure S1D**. Comparison of the two tools for peak calling is described in the main text (lines 94-97), and in the methods section (Data analysis paragraph).

3. Line 38. Validate the anti-G4 antibody and/or provide convenient references.

Each used batch of purified BG4 antibody was validated *in vitro* by mean of native electrophoretic mobility shift assay (EMSA). A dose dependent binding intensity was measure for BG4 when incubated in the presence of three human genome G4-folded sequences (c-myc, bcl.2 and c.kit), while no binding was detected in the presence of a G-rich non G4 single stranded sequence (scrambled). One representative EMSA experiment has been added in **Supplementary Figure 1A** and described in the material and methods section as well as in the main text: "BG4 antibody was purified and its capacity to discriminate for G4 structures over non folded single strand sequences was in vitro validated by EMSA assay."

References of papers employing the same strategy and obtaining comparable results in BG4 activity validation were mentioned in the main text: doi: 10.1038NCHEMBIO.2228, doi: 10.1371/journal.pone.0158794

4. Line 42. Use of "significant". Please confirm that this is accompanied by appropriate stats throughout the manuscript.

The use of the term "significant" has been now supported by statistics throughout the manuscript:

- Line 95, Fig S1C: the significance level of CG and AT content in BG4 ChIP peaks with respect to their abundance in the whole human genome has been calculated with a T-Test and asterisk indicating p-value have been plotted and described.
- Line 118, Fig 1C: the significance level of transcript amount (TPM) of each G4 category was calculated with T-Test with respect to the no G4 group. P-values were indicated in the graph, and described in the Figure legend, as well as mentioned in text at line 118.
- Line 219, statistical significance for differentially expressed genes in 93T449 and HaCaT cell lines was calculated by mean of apegm and DESeq2 R packages considering s-value s-value < 0.1; fold change > 1.0 as was indicated in the materials section. For clarity, this has been now indicated also in the main text.
- Supplementary Figure 5. Significance levels have been calculated by T-test, described and discussed in figure legend and main text results (line 140 p. 6).
- Significance of overrepresented TFs present in G4-ChIP peaks was calculated by mean of Homer software and p-values have now been specified in main text, Figure4A legend and discussion section.

5. Line 43. High GC frequency "towards" the ChIP peak center but not "at" the center for G and C.

We now specified in the main text: "*the highest GC frequency increasing towards the ChIP peak centre, with a maximum GC occurrence at about 50 bp from the peak centre.*"

6. Line 128 (SF6?). Correlating the presence of G4s with a parameter like cell growth, which integrates many processes, might not be relevant, unless experimentally demonstrated independently.

We agree with the referee, we have removed these data.

7. Lines 161-162. This claim goes way too far from being supported by the results. This may require deleting G4s and determine effect on transcription at some individual loci.

We made the claim less strong, to clearly state that it is an hypothesis that we derived from the results: "*Together, these data led us to the hypothesis that different cell types modulate G4 folding to establish or maintain their transcriptional program.*"

8. Correlation with TF binding sites may be related to the high GC content of those sites, not necessarily to the function of these TFs. Please comment.

The correlation between TF binding sites and folded G4s is not dependent from the GC-content of those regions. Evidence that supports the independence between TF binding and GC-content is: i) the absence of GC-richness in the AP-1 consensus binding sequence ATGAGTCA (**see Figure 5A**), which indicates the absence of artefact when correlating G4 formation and AP-1 recruitment at its binding site. In the case of AP-1, this statement is also supported by the fact that its binding site is strongly overrepresented at the peak centre of ChIP-G4s (**Figure 5B**), while the highest GC frequency within the G4 peak, is observed at about 50 bp far from the peak center (**Supplementary Figure 1D**), further supporting the absence of correlation between intrinsic GC-richness and AP-1 binding site. Less obvious is the case of Sp1, due to the intrinsic GC-richness of its consensus sequence (GGGGCGGGG), which could participate itself to G4 folding. To clarify this issue, in **Figure 5C** we showed that, despite the actual binding sites for both TFs are more frequent in concomitance with sequences that potentially form G4 (oG4 and pG4, with respect to the other genomic regions), there is a clear enrichment in the actual

binding sites for both AP-1 and Sp1 in those regions that fold into G4s in cells (ChIP-G4). This observation is a strong indication that the simple GC-richness or G4 folding potential do not determine AP-1 and Sp1 binding, but rather that the presence of folded G4s is a preferential condition for TF binding.

This point has now been more explicitly discussed in the results section at pages 14-15: *"This result also indicates that is not the simple GC-richness or G4 folding potential to determine the binding of AP-1 and Sp1, but the presence of folded G4 structures is a preferential condition for having TF binding. Another supporting observation that the correlation between TFs binding and G4 presence is not an artefact due to the GC-richness of such regions, comes from the non-prominent GC frequency of AP-1 consensus sequence (ATGAGTCA). Moreover, AP-1 binding site is strongly overrepresented at the peak centre of ChIP-G4s (Figure 5B), while the highest GC frequency within the G4 peak, is observed at about 50 bp far from the peak centre, (Supplementary Figure 1D) supporting the absence of correlation between intrinsic GC-richness and AP-1 binding site. Less obvious is the case of Sp1, due to the intrinsic GC-richness of its consensus sequence (GGGGCGGG), which could participate itself to G4 folding."*

And in discussion:

"Moreover, by comparing the actual TFBS for AP-1 and Sp1 (ENCODE database) with G4 permissive regions, we highlighted that the correlation between TFs binding and G4s is not dependent on the intrinsic GC-richness of those sequences, but that the presence of folded G4 structure drives the correlation with TFs binding."

REVIEWERS' COMMENTS

Reviewer #1 (Remarks to the Author):

The authors have addressed the concerned I have raised and I am now supportive of publication.

Reviewer #2 (Remarks to the Author):

The manuscript entitled "Promoter G-quadruplexes and transcription factors cooperate to shape the cell type-specific transcriptome" by Lago et al. strengthens and generalizes the notion of G-quadruplexes as determinants of the cell transcription program through cell differentiation. Genome-wide seq approaches and co-immunoprecipitation data with transcription factors corroborate previous findings on keratocytes and expand the concept to other cell lines, specifically to HaCaT and 93T449 lines. The authors exhaustively addressed all the concerns raised upon revision, satisfactorily strengthening their hypothesis and experimental design by providing convincing additional data. I therefore deem to support the manuscript for publication on Nature Communication upon minor adjustment. I recommend to amend the abstract to better reflect the manuscript content its current form and thorough proofreading.

Reviewer #3 (Remarks to the Author):

The authors have addressed my concerns satisfactorily.

Reviewer #1 (Remarks to the Author):

The authors have addressed the concerned I have raised and I am now supportive of publication.

We thank the reviewer for this positive evaluation

Reviewer #2 (Remarks to the Author):

The manuscript entitled "Promoter G-quadruplexes and transcription factors cooperate to shape the cell type-specific transcriptome" by Lago et al. strengthens and generalizes the notion of G-quadruplexes as determinants of the cell transcription program through cell differentiation. Genome-wide seq approaches and co-immunoprecipitation data with transcription factors corroborate previous findings on keratocytes and expand the concept to other cell lines, specifically to HaCaT and 93T449 lines. The authors exhaustively addressed all the concerns raised upon revision, satisfactorily strengthening their hypothesis and experimental design by providing convincing additional data. I therefore deem to support the manuscript for publication on Nature Communication upon minor adjustment. I recommend to amend the abstract to better reflect the manuscript content its current form and thorough proofreading.

We thank the reviewer for this positive evaluation. We have modified the abstract to include the modifications included during the revision. We have also extensively proofread the manuscript.

Reviewer #3 (Remarks to the Author):

The authors have addressed my concerns satisfactorily.

We thank the reviewer for this positive evaluation